# Temperature modulates stress response in mainstream anammox reactors

Robert Niederdorfer [1]✉, Damian Hausherr [2], Alejandro Palomo [3], Jing Wei[4], Paul Magyar [5], Barth F. Smets [3], Adriano Joss[2] & Helmut Bürgmann [1]

Autotrophic nitrogen removal by anaerobic ammonium oxidizing (anammox) bacteria is an energy-efficient nitrogen removal process in wastewater treatment. However, full-scale deployment under mainstream conditions remains challenging for practitioners due to the high stress susceptibility of anammox bacteria towards fluctuations in dissolved oxygen (DO) and temperature. Here, we investigated the response of microbial biofilms with verified anammox activity to DO shocks under 20 °C and 14 °C. While pulse disturbances of 0.3 mg $L^{-1}$ DO prompted only moderate declines in the $NH_4^+$ removal rates, 1.0 mg $L^{-1}$ DO led to complete but reversible inhibition of the $NH_4^+$ removal activity in all reactors. Genome-centric metagenomics and metatranscriptomics were used to investigate the stress response on various biological levels. We show that temperature regime and strength of DO perturbations induced divergent responses from the process level down to the transcriptional profile of individual taxa. Community-wide gene expression differed significantly depending on the temperature regime in all reactors, and we found a noticeable impact of DO disturbances on genes involved in transcription, translation, replication and posttranslational modification at 20 °C but not 14 °C. Genome-centric analysis revealed that different anammox species and other key biofilm taxa differed in their transcriptional responses to distinct temperature regimes and DO disturbances.

[1] Eawag, Swiss Federal Institute for Aquatic Science and Technology, Department of Surface Waters-Research and Management, 6047 Kastanienbaum, Switzerland. [2] Eawag, Swiss Federal Institute for Aquatic Science and Technology, Department of Process Engineering, 8600 Duebendorf, Switzerland. [3] Department of Environmental Engineering, Technical University of Denmark, Kgs Lyngby, Denmark. [4] Empa, Swiss Federal Laboratories for Materials Science and Technology, Laboratory for Air Pollution & Environmental Technology, 8600 Duebendorf, Switzerland. [5] Department of Environmental Sciences, University of Basel, Basel, Switzerland. ✉email: robert.niederdorfer@eawag.ch

utotrophic nitrogen removal by biological anaerobic ammonium oxidation (anammox) is increasingly implemented as an energy-efficient mechanism of fixed nitrogen elimination during wastewater treatment. Deployed for mainstream wastewater treatment plants (WWTPs), it may even permit operation under energy autarky[1–3]. In contrast to conventional nitrification–denitrification system, external carbon sources are not required to reach very low effluent nitrogen concentration, less aeration is needed, and most of the organic load can be diverted for valorization, e.g. biogas or bio-plastics production[4–6]. Autotrophic N removal with anammox involves the simultaneous oxidation of ammonium ($NH_4^+$) and reduction of nitrite ($NO_2^-$) under oxygen-free conditions[7]. In engineered systems, aerobic ammonia-oxidizing bacteria (AOB) oxidize a fraction of the available $NH_4^+$ to $NO_2^-$ (nitritation), which is subsequently used as a terminal electron acceptor by anammox bacteria to oxidize the remaining $NH_4^+$ to $N_2$ (ref. [8]). The process can be realized in single-stage[9] or two-stage bioreactor systems[10]. Currently, autotrophic N removal with anammox coupled to nitritation is already widely applied and represents a robust method for the treatment of wastewaters with high N concentrations under mesophilic conditions, e.g. effluents from anaerobic sludge digestion[2,11]. However, development of stable anammox processes for mainstream municipal WWTPs suffers from unexplained process instabilities due to unexpected fluctuation of environmental temperature and dissolved oxygen (DO)[12–14].

Anammox bacteria are characterized by very slow growth rates, low cell yields[15], and a high sensitivity to changing environmental conditions[16]. For their application in wastewater treatment, they are grown either in biofilm reactors on various carrier materials or as granules to retain sufficient biomass[17]. All respective reactor configurations support the formation of complex microbial communities[18] with many potential synergistic and antagonistic interactions[19,20]. Anammox, nitrification, heterotrophic denitrification, ammonification, as well as N incorporation may occur simultaneously within the system[21,22], thus complicating efforts to disentangle the sources of process instabilities.

Several studies have identified environmental stressors that affect the anammox process on the performance level (reviewed in Jin et al.[23]). Transient pulses of common wastewater constituents in the influent, such as heavy metals, phosphates, high $NO_2^-$ concentration, and sulfides have been reported as problematic factors, causing process instabilities[12,13,24]. However, DO and temperature were frequently discussed as the most critical factors with regards to stable operation of the anammox process[12,25–28]. Low temperature is a seasonal stressor especially for mainstream anammox[29]. In contrast, even short-term oxygenation of anammox systems has been associated with system failures[28]. Oxygen control is of particular importance when it comes to the operation of the PN/A process in a single reactor. Numerous studies implementing intermittent aeration strategies have however demonstrated that successful nitritation and anammox processes are possible under these conditions[30,31].

So far, the focus of these studies was to investigate the effect of applied oxygen or temperature disturbances on the process level and AMX efficiency, with a few studies providing further information on the microbial community composition[14,26,32]. However, the stability problems have so far not been resolved, and a mechanistic understanding of the events that have led up to and potentially caused process failures has not yet emerged. This provides a motivation for studies using multi-omics approaches to understand what happens on the molecular level during performance failures and/or the ensuing recovery processes. Metagenomic shotgun sequencing allows to gain a comprehensive insight in the functional potential of communities involved in the anammox process[19,20,33,34] but is by itself unlikely to yield useful information on reversible short-term disturbance effects, which, in most of the cases, are not leading to shifts in the microbial community composition. In contrast, Metatranscriptomics provides information on cellular activities on short time scales. Metatranscriptomic sequencing has already demonstrated its potential to provide detailed insights into the differences in gene expression of prokaryotic key players in AMX reactors under sidestream conditions[19,35]. Here we use, based on these pioneering studies, for the first time metatranscriptomics to investigate stress response in AMX reactors. In the framework of a fully replicated experimental design, we combine a multi-omics approach with a focus on metatranscriptomics with biochemical measurements. Understanding the transcriptional response of microbial communities but also its individual members to stress and how this response links to process-level effects is necessary for understanding process stability. A better understanding of the processes underlying system failure or stability are crucial for successful implementation of autotrophic N removal at full scale.

Specifically, we investigated the performance dynamics of complex anammox biofilms in response to short-term DO perturbations under two different temperature regimes. We hypothesized (1) that anammox biofilms already exposed to certain stressors (e.g. lower temperatures) would experience stronger and longer lasting disturbance of the anammox process by the applied DO shocks. We further hypothesized (2) that DO shocks would result in a specific transcriptional response of the biofilm community in general, and anammox bacteria specifically. By analyzing the transcriptional response of different species, we aim to obtain information on the stress level experienced by different microorganisms. To verify the reproducibility of observed effects, we ran parallel disturbance experiments in three laboratory-scale sequencing batch reactors (SBR) under comparable conditions. By combining high-resolution monitoring of performance parameters with omics-based analysis of community composition, biodiversity, and gene transcription, we aimed to determine transcriptional mechanisms connected to process disruptions. By mapping metatranscriptome data to metagenome-assembled genomes of anammox bacteria and other major biofilm members in a bioreactor system for autotrophic N removal, this study demonstrates that the impact of disturbances can be determined down to the level of transcriptional activity in individual microbial species using meta-omics tools.

## Results

**Process level performance and disturbance response**. Per-cycle reactor performance was comparable in all three reactors during the 3 days of operation under baseline conditions under both temperature regimes (Fig. 1B, C). We did not observe anomalies in reactor activity potentially imposed by the carrier transfer from the pilot reactor (17 °C) to the small volume bioreactors (14 °C or 20 °C). The average $NH_4^+$ removal rates per hour within an SBR cycle during baseline conditions were $3.7 \pm 0.5$ $mg_{NH4-N}$ $L^{-1}$ at 20 °C and $3.2 \pm 0.4$ $mg_{NH4-N}$ $L^{-1}$ at 14 °C, respectively. However, the $NH_4^+$ removal rates of R1 (Fig. 1B, C) were significantly higher compared to the other reactors independent of the temperature regime ($p < 0.05$, Student's $t$-test). The average SBR cycle duration during the experiments was $7.2 \pm 2.0$ h at 20 °C and $8.8 \pm 2.3$ h at 14 °C, respectively. $NH_4^+$ removal rates in all reactors experienced similar trends over the course of the experiment. While at 20 °C the average $NH_4^+$ removal per hour within a cycle tended to increase after the baseline period ($3.9 \pm 0.7$ $mg_{NH4-N}$ $L^{-1}$, not significant), the reactor performance showed an opposite, decreasing trend at 14 °C to an average of $2.5 \pm 0.4$ $mg_{NH4-N}$ $L^{-1}$ which was significant (linear regression, $p < 0.005$ for all reactors; R1: $r^2 = 0.8$; R2: $r^2 = 0.7$; R3: $r^2 = 0.6$)

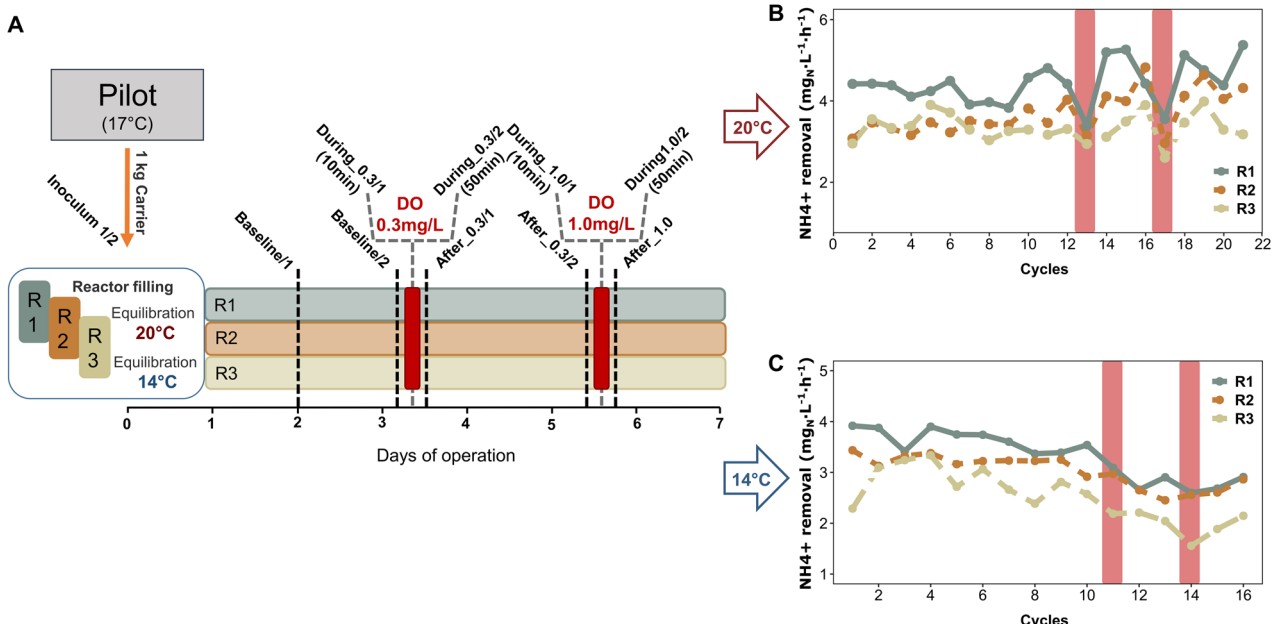

**Fig. 1 Sampling scheme and average NH$_4^+$ removal rate. A** Experimental setup and carrier sampling scheme with sample names. **B** Average NH$_4^+$ removal per SBR cycle in mg Nitrogen/Liter/hour during the 20 °C experiment. Colors of the linecharts denote the different reactors (green: Reactor1; brown: Reactor2; beige: Reactor3). Red bars highlight the DO disturbed cycles. **C** Average NH$_4^+$ removal per SBR cycle mg Nitrogen/Liter/Hour during the 14 °C experiment. Colors of the lines denote the different reactors. Red bars highlight the DO disturbed cycles.

(Fig. 1B, C). This explains the prolonged SBR cycles during the 14 °C treatment, as cycle duration was directly dependent on the NH$_4^+$ removal efficiency.

The DO perturbations caused an obvious impact on the average per cycle NH$_4^+$ removal rate at 20 °C, while the DO shocks at 14 °C did not impact the per cycle activity (Fig. 1B, C red bars). To understand the susceptibility of reactor performances to DO perturbations in more detail, we analyzed the SBR cycles and corresponding average NH$_4^+$ removal rates at a half-hour temporal resolution (Fig. 2A, B). Here, a clear distinction in performance decay between high and low oxygen stress was observed. Pulse disturbances of 0.3 mg L$^{-1}$ DO prompted only moderate declines in the average NH$_4^+$ removal rates (~40% maximum activity loss) under both temperature regimes. On the other hand, 1.0 mg L$^{-1}$ of DO completely inhibited NH$_4^+$ removal activity in all reactors at 20 °C (Fig. 2A) but only partly at 14 °C (Fig. 2B). However, after the 1.0 mg L$^{-1}$ DO perturbation, all disturbance effects were rapidly reversible, even within the perturbed SBR cycle, except for R3 during the 14 °C experiment. The observed recover time to baseline performance averaged 34.7 ± 4 min at 20 °C and 40.6 ± 8 min at 14 °C, respectively (Supplementary Fig. 1A). Interestingly, the time to reach maximum inhibition of the NH$_4^+$ removal rate was nearly the same for both temperature regimes and oxygen concentrations (Supplementary Fig. 1B). On average, it took 72 ± 4 min to reach the maximum inhibition of the NH$_4^+$ removal rate after start of the DO perturbation.

**Microbial community structure and genetic potential.** Based on the short-term nature of the experiment and generally weak disturbance effects, we did not expect changes of the microbial community composition. Our metagenome analysis confirmed that microbial community composition and functional potential (functional gene pool of the community) was stable (Fig. 3 and Supplementary Figs. 2, 3). Differences between the samples,

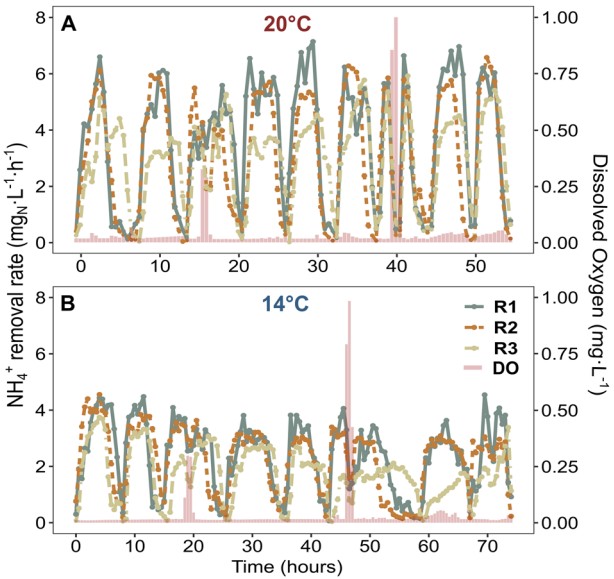

**Fig. 2 Process level performance. A** NH$_4^+$ removal rates per hour over nine SBR cycles (20 °C). x-Axis denotes the time in hours, the y-axis (left) displays the NH$_4^+$ removal rate in mg Nitrogen/Liter/hour. Colors of the linecharts denote the different reactors (green: Reactor1; brown: Reactor2; beige: Reactor3). Concentrations of dissolved oxgen (red bars) in mg DO/Liter are shown on the second y-axis (right). **B** NH$_4^+$ removal rates per hour over nine SBR cycles (14 °C).

mainly on the species level, are thought to primarily reflect carrier-to-carrier variation[36].

The microbial community in the biofilm carriers consisted of a diverse assemblage of taxa, many of which are typical for mainstream anammox WWTP[19,20,33]. Members of *Proteobacteria*, *Chloroflexi*, *Planctomycetes*, and *Actinobacteria* dominated the

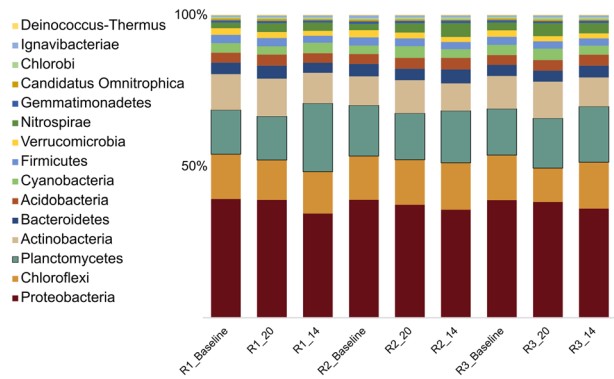

**Fig. 3 Microbial community composition.** Relative abundance of the top 15 bacterial phyla of the biofilm carrier community derived from taxonomic assignment of metagenomic sequence reads using Kaiju. Labels denote Reactor (R1, R2, R3) and time of sampling, _Baseline (Pooled from 20 °C and 14 °C experiment), _20 (after the 20 °C experiment), _14 (after the 14 °C experiment). Colors represent the different bacterial phyla.

biofilms, accounting for ~70% of the total community composition, However, *Proteobacteria* represented the largest Phylum of the community accounting for ~40% of the total abundance (Fig. 3). Other organisms that displayed moderate abundance were affiliated with *Nitrospirae*, *Gemmatimonadetes*, and *Acidobacteria*.

Genera with known importance to the N cycle, e.g. anammox bacteria, denitrifying bacteria, and nitrite oxidizing bacteria (NOB), were abundant. *Candidatus Brocadia* represented the most abundant genus of the community (Supplementary Fig. 2). Interestingly, NOB from the Genus *Nitrospira* also accounted for ~3% of the total community. In contrast, only a very small fraction of the community (~0.4%) could be assigned to genera of aerobic AOB.

As expected, characterization of functional capacity of the community from contig gene content revealed that genes involved in cluster of orthologs (COG) energy production and conversion (C), protein synthesis (J, K, L), amino acid metabolism (E), and signal transduction (T) were the most abundant (summed metagenomic read abundances in Genes per million (GPM)) (Supplementary Fig. 3). For specific functional subsystems, we found that genes involved in the carbohydrate metabolism and production of vitamins and cofactors comprised the most abundant fractions. Genes involved in N transformations did not occur in high abundance (Supplementary Fig. 4). However, genes belonging to the anammox metabolism (reddish colors, Supplementary Fig. 5) such as hydrazine dehydrogenase (*hdh*) and the hydrazine synthase cluster (*hzsαβγ*) and denitrification (bluish colors) were the most abundant ones within the N cycle category. To our surprise, also genes involved in dissimilatory nitrate reduction to ammonium (DNRA) occurred in moderate abundance, while assimilatory $NO_3^-$ reduction genes were only present in small numbers (Supplementary Fig. 5).

Since no major shifts were observed in the community level nor in the functional potential between the nine metagenomic samples, a co-assembly was applied to ensure higher coverage for further transcriptomic analysis.

**Identity and abundance of Metagenome Assembled Genomes.** To be able to dissect the ecology and stress response of individual bacterial lineages from the complex microbial communities used in these experiments, the recovery of Metagenome Assembled Genomes (MAGs) is a necessary prerequisite. We investigated 12 MAGs ranging from medium to high quality (>80% completeness, <10% contamination)[37] from the co-assembled metagenomic libraries. These MAGs were affiliated with the phyla *Planctomycetes* (AMX),

*Nitrospirae* (NOB), *Chloroflexi* (CFX), *Proteobacteria* (APT), *Bacteroidetes* (BCD), and *Gemmatimonadetes* (GMT) (Table 1).

The average relative abundance of the MAGs was calculated from the amount of total filtered DNA reads from each of nine metagenome samples that mapped to all retrieved MAGs. On average, the two most abundant MAGs accounted for 13.9% and 4%, of the community, respectively (Table 1). The other MAG's abundances ranged between 2.0 and 0.5%.

A phylogenetic tree based on the protein sequences of 37 single-copy marker genes[38] (Supplementary Fig. 6) showed that the majority of retrieved MAGs displayed high similarities to reference genomes from previous bioreactor studies[19,20,33]. Our results suggest that the six recovered putative anammox genomes (AMX1-6) are all representatives of the *Brocadia* genus and both NOB MAGs were closely related to *Nitrospira defluvii*. AMX4 and AMX2 created their own branch within the Genus *Brocadia*. The MAG affiliated with phylum Chloroflexi was assigned to the genus *Candidatus* Promineofilum, a well-known filamentous member of activated sludge communities with a facultative anaerobic lifestyle in WWTP[39].

The taxonomic classification and relative abundance is also in line with our findings on the metagenome level, where anammox bacteria accounted for ~20% of the community (Fig. 3).

**MAGs and the N-cycle.** The recovered MAGs were analyzed with regard to their functional role in the bioreactors biological N-cycle network. As expected, all anammox MAGs carried at least two of the most important anammox genes, *hdh* and hydroxylamine oxidoreductase (*hao*) (Supplementary Table 1). However, *hzs* could not be annotated in AMX4 and AMX5, and may have been lost in the assembly. Manual BLAST analysis of the contigs of these MAGs confirmed the absence of *hzs* homologs. Interestingly, except for AMX3, none of the anammox MAGs contained homologs of *nirK* or *nirS* genes, which are typically responsible for the first step of the anammox process[40,41]. The inconsistent presence of nirS/nirK or neither throughout the anammox clades have led to the hypothesis that one of the hao-like enzymes conserved across all anammox genera could be the physiological nitrite reductase[40]. The majority of retrieved anammox MAGs also contained *nrfH* genes (Supplementary Table 1), which encodes for enzymes responsible for the reduction of $NO_2^-$ to $NH_4^+$. We were not able to annotate Nitrite:Nitrate Oxidoreductase (*nxr*) in the anammox MAGs perhaps due to its high sequence similarity to bacterial nitrate reductases[42].

All other MAGs encoded capabilities for partial or full denitrification, DNRA and assimilatory $NO_3^-$ reduction (Supplementary Table 1). The first step in denitrification appears to be orchestrated by GTD, and BCD carry the genes for respiratory nitrate reductase (*narGHI*) that reduces $NO_3^-$ to $NO_2^-$. The ability to reduce $NO_2^-$ to nitric oxide (NO) (*nirS*, *nirK*) was found in the Nitrospira (NOB1, NOB2), Chloroflexi, and the Proteobacterium (APT) genomes. All of them contain either one or both nitrite reductase variants. BCD and all anammox genomes have the functional competence to reduce NO to nitrous oxide ($N_2O$) via the nitric oxide reductase (*norB*, *norC*) to conclude the next step of partial denitrification. Finally, CFX, APT, GTD, and BCD expressed genes for the reduction of $N_2O$ to $N_2$ via nitrous-oxide reductase (*nosZ*). Thus, all retrieved MAGs seem to metabolically participate in the N cycle and are important for the overall N removal in this ecosystem.

**Community-wide transcriptomic analysis.** Metatranscriptomics allowed us to assess impacts of temperature regime and disturbance events on the community-wide transcriptional activity. RNAseq yielded on average $2 \times 10^7$ reads per sample after quality

**Table 1 Binning statistics of MAGs recovered from the anammox community.**

| MAG ID | Taxonomy | Relative abundance | Completeness (%) | Contamination (%) | GC (%) | N50 (scaffolds) | Genome size (bp) | # contigs |
|---|---|---|---|---|---|---|---|---|
| AMX1 | Planctomycetes; Planctomycetia;Candidatus Brocadiales;Candidatus Brocadiaceae | 13.701 | 80.12 | 6.55 | 43.7 | 2073 | 2985008 | 1485 |
| AMX2 | Planctomycetes; Planctomycetia;Candidatus Brocadiales;Candidatus Brocadiaceae | 3.940 | 94.5 | 2.75 | 41.8 | 12593 | 3160181 | 338 |
| AMX3 | Planctomycetes; Planctomycetia;Candidatus Brocadiales;Candidatus Brocadiaceae | 1.130 | 83.17 | 4.20 | 43.3 | 4144 | 3120508 | 928 |
| AMX4 | Planctomycetes; Planctomycetia;Candidatus Brocadiales;Candidatus Brocadiaceae | 0.687 | 85.71 | 0.55 | 42.1 | 23036 | 2898609 | 173 |
| AMX5 | Planctomycetes; Planctomycetia;Candidatus Brocadiales;Candidatus Brocadiaceae | 0.464 | 94.44 | 1.648 | 42.1 | 31411 | 2844886 | 138 |
| AMX6 | Planctomycetes; Planctomycetia;Candidatus Brocadiales;Candidatus Brocadiaceae | 0.438 | 91.2 | 4.945 | 44 | 21519 | 3360983 | 214 |
| APT | Proteobacteria; Alphaproteobacteria; Rhizobiales; Bradyrhizobiaceae | 0.739 | 99.49 | 1.418 | 63.9 | 167385 | 5912068 | 52 |
| BCD | Bacteroidetes;Flavobacteriia; Flavobacteriales; Flavobacteriaceae | 0.473 | 100 | 0.358 | 32.7 | 88353 | 3658476 | 64 |
| CFX | Chloroflexi;Ardenticatenia; Ardenticatenales; Ardenticatenaceae | 1.959 | 91.09 | 2.727 | 60.7 | 27429 | 4045426 | 248 |
| GTD | Gemmatimonadetes; Gemmatimonadetes; Gemmatimonadales; Gemmatimonadaceae | 0.477 | 98.9 | 2.197 | 64.4 | 1505125 | 4063306 | 58 |
| NOB1 | Nitrospirae;Nitrospira; Nitrospirales;Nitrospiraceae | 1.066 | 89.48 | 1.818 | 59.7 | 119970 | 3821942 | 73 |
| NOB2 | Nitrospirae;Nitrospira; Nitrospirales;Nitrospiraceae; Nitrospira;Nitrospira defluvii | 1.029 | 81.25 | 6.666 | 59.5 | 9669 | 2882492 | 351 |

filtering. Reads were mapped against the metagenomic co-assembly derived from the nine metagenomic samples to provide insight into the transcriptional response of the whole community. The majority of genes, displaying the highest fluctuations in transcript abundance, could only be annotated as hypothetical proteins, therefore we decided to compare the changes in transcript abundance on the community level. The community-wide transcription differed significantly between the two temperature

regimes and significantly between the reactors during the 20 °C experiment (PERMANOVA: $p < 0.005$).

The non-metric multi dimensional scaling (nMDS) ordination based on global transcript abundances (Fig. 4) displays, independent of the reactor, a tight clustering of the 14 °C samples, indicating only minor changes in the transcriptional status of the community in response to the applied DO disturbances. On the other hand, samples from the 20 °C experiment are dispersed in the ordination, suggesting a larger transcriptional variance during the 20 °C experiment. The dissimilarities between the two inoculum samples (black triangles), taken 1 week apart at the exact same time during an SBR cycle, indicate that the inoculum was generally stable, but also that variability in transcriptomic data has to be expected even without experimental intervention—either due to the dynamic nature of this engineered ecosystem or due to methodological error. These differences had no effect on the process level, as we did not observe significant differences in $NH_4^+$ removal rates between the two fillings of the experimental reactors ($p < 0.05$, Student's $t$-test).

DO disturbances did not induce a consistent community-wide transcriptional response at either temperature regime, i.e. samples obtained at "stress" conditions did not cluster consistently apart from baseline samples (Supplementary Fig. 7).

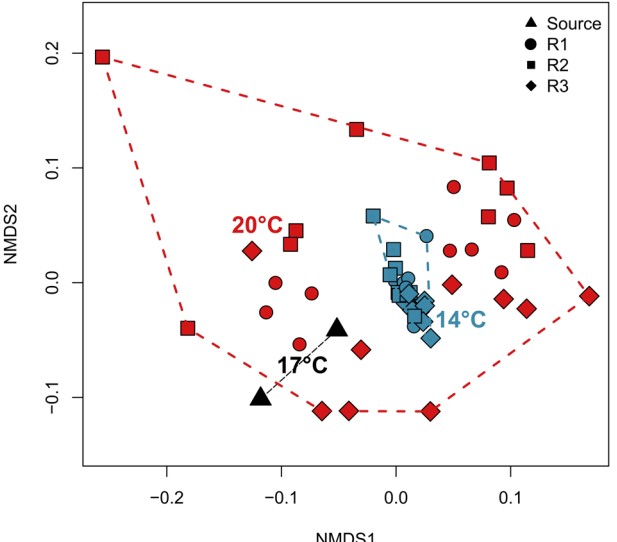

**Fig. 4 Transcriptomic differences on the community level.** nMDS analysis based on Jaccard dissimilarities depicting overall gene expression during the time-points of the experiment under different temperatures. Colors of clusters denote the respective temperature of the experiment (black: Inoculum, red: 20 °C, blue: 14 °C). Symbols respresent the different reactors. Reactor1 is missing one Baseline sample of the 14 °C experiment due to insufficient coverage of the metatranscriptome. Stress value: 0.09.

**Transcriptional response of functional gene categories.** To explore further implications of temperature on transcript abundance and to assess if oxygen affected transcription of specific gene networks, we investigated the influence of disturbances on gene transcription on the level of functional categories (COG) and for N metabolism, respectively.

Transcript abundance was, unsurprisingly, the highest for genes in the COG categories C (Energy production and conversion), E (Amino acid transport and metabolism), J (Translation), K (Transcription), and O (posttranslational modification) independent of the temperature regime (Fig. 5A). In contrast to our findings on the global transcriptomic profile,

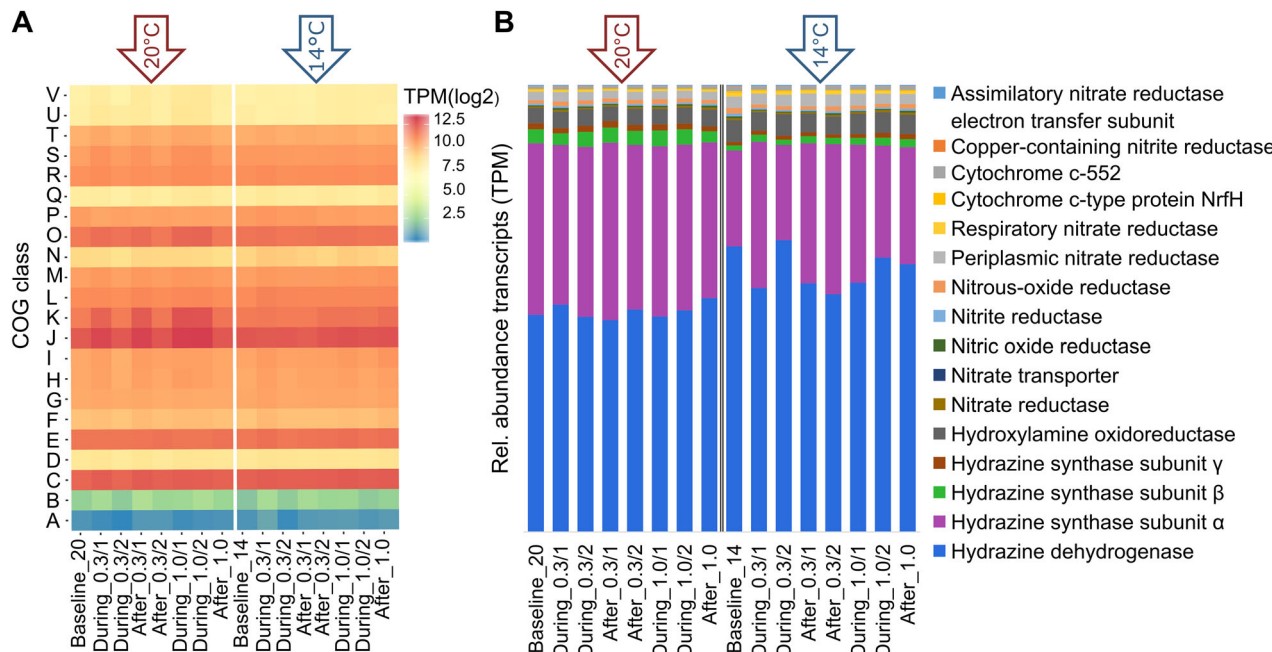

**Fig. 5 Transcriptomic differences on functional level. A** Heatmap based on the cumulated transcripts per million (log2) from all genes falling into certain cluster of orthologs (COG) categories ($y$-axis). Time-points ($x$-axis) are based on the averaged reactors ($n = 3$). **B** Relative abundance of transcripts in TPM of genes involved in the nitrogen cycle. The left part shows the 20 °C samples, the right part the 14 °C experiment, as indicated by the arrows.

DO disturbances during the 20 °C regime appeared to have notable impacts on some COG categories. Especially, genes within the clusters J, K, and O displayed elevated relative transcription during the first 10 min of 0.3 mg L$^{-1}$ DO (15 ± 3% increase), after the 0.3 mg L$^{-1}$ DO disturbance (15 ± 6% increase), and during the whole period of the 1.0 mg L$^{-1}$ DO perturbation (26 ± 7% and 26 ± 4% increase, respectively). The most prominent genes within these categories were annotated as cold shock-like proteins (K) and a variety of ribosomal proteins, RNA polymerase sigma factors (J), and Heat-shock proteins (O). These genes displayed high fluctuations in relative transcription at 20 °C but maintained a stable transcription over the whole experiment at 14 °C. In line with findings for global gene transcription (Fig. 4), changes in transcription of COG classes were in general much less pronounced at 14 °C. However, transcript abundances during baseline conditions were comparable independent of the temperature ($p > 0.5$, Student's $t$-test).

Transcript abundance confirmed the dominance of the anammox process already seen in the metagenome. Highest transcription was observed for genes involved in the anammox metabolism (Fig. 5B). Notably, transcripts of hydrazine dehydrogenase and hydrazine synthase subunit alpha had up to twofold higher relative transcript abundance than other N cycle genes (Fig. 5B). This is in line with previous findings on AMX genomes and corresponding transcript abundances[19,34].

Taking all N cycle genes into account, temperature again had an distinct effect on the transcriptional status of the community (see ordination in Supplementary Fig. 8). While transcription of $hdh$ genes was elevated at 14 °C, the transcription of $hzs$ subunits α and β were significantly reduced ($p < 0.05$, Student's $t$-test) (Fig. 5B). In contrast to the community-level transcription trend, $hdh$ displayed constant levels of transcription at 20 °C, but more variations at 14 °C. Similar to recent reports[19], we also found moderate expression levels for partial or full denitrification. Genes encoding for denitrification and DNRA did not display significant differences between the temperature regimes ($p < 0.05$, Student's $t$-test).

**Transcriptional response of individual MAGs.** Changes in the transcription at the level of functional genes in a microbial community as described above may arise from global changes of transcriptional activity in many different species as well as from specific transcriptional regulation within individual species. To investigate this, we combined MAG reconstruction with our transcriptomic approach.

Relative transcript abundances as a proxy for expression levels were calculated from the total filtered mRNA reads from 56 RNA samples, which were individually mapped against the recovered MAGs, normalized for respective genome length and sequencing depth and expressed as transcripts per million (TPM) (Supplementary Table 2)[19,43]. The two most abundant anammox MAGs (~18% relative abundance) together were significantly ($p < 0.05$, Student's $t$-test) responsible for the highest transcriptional activity during the 20 °C and 14 °C regimes, respectively (Fig. 6A). Less abundant *Candidatus* Brocadia MAGs (AMX3, AMX4, AMX5, and AMX6) demonstrated only moderate transcriptional activity. Temperature had the strongest effect on AMX5, which was more active at 14 °C while AMX2 appeared more active at 20 °C. To highlight the relationship between transcriptional activity and abundance, we calculated the ratio between these two metrics for each anammox MAG. Ratios were comparable during the different temperature regimes except for AMX2 and AMX5. The high ratio of AMX2 can be attributed to a low abundance but high transcriptional activity.

After finding these differences in the transcriptional activity of the retrieved MAGs, we compared transcriptional patterns within

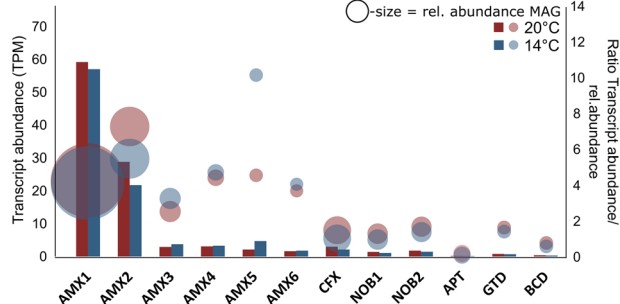

**Fig. 6 Relative abundance and transcription of MAGs.** Barcharts representing the relative transcript abundance (in TPM) of MAGs recovered from the bioreactor. Bubbles represent the ratio between transcript abundance and corresponding relative abundance for each MAG. The second $y$-axis denotes the ratio levels. The size of bubbles correspond to the relative abundance of the MAGs. Supplementary Table 2 gives more details on mapping results.

the functional classifications of COGs. Here, we summed up the TPM of all genes in each COG class for each MAG at each time-point. This dataset was investigated by ordination. For ease of interpretation, we split the nMDS (Jaccard dissimilarity) based on the temperature regimes (Fig. 7). Transcriptional activity of COG categories shows MAGs to be clearly and consistently distinct, as evidenced by clusters within the nMDS space, indicating that the members of the same functional groups (e.g. AMX and NOB, which are in our case also from the same genus) were functionally distinct at the COG level. MAGs from different functional groups clustered more distant (hull polygons in Fig. 7). Importantly, we found the degree of transcriptional change over time differed between individual MAGs (spread of points for each MAG in the nMDS space). Especially the spread between time-points within the NOB cluster suggested dynamic changes in gene transcription independent of the temperature regime. Some MAGs displayed a clear temperature dependency in their transcriptional responses over the course of the experiment (Supplementary Fig. 9). Specifically, within the AMX cluster, we found that temperature strongly affected the transcriptional profile of AMX3, AMX5, and AMX6, as well as the degree of transcriptional change (Fig. 7 and Supplementary Fig. 9).

## Discussion

Using laboratory-scale bioreactors and meta-omic approaches, we were able to investigate anammox process stability toward DO pulse disturbances and different temperature regimes on various ecosystem and biological levels.

While both temperature regimes started with similar N removal efficiencies, the lower temperature treatment had a significant longer-term negative effect on anammox performance. In agreement with previous findings on temperature-induced performance loss in anammox reactors[25,44,45], this highlights again that one of the main issues with the mainstream application of autotrophic N removal is the temperature sensitivity of anammox bacteria. DO disturbances had a clear, but short-term, reversible influence on the anammox process level. In strong contrast to our expectations, DO disturbances had a much more severe impact at 20 °C on the $NH_4^+$ removal rate, resulting in, depending on the DO concentration, partial to even full inhibition. During the 14 °C experiment, DO shocks did not lead to these drastic decreases in anammox activity (Fig. 2).

Looking beyond the black box of performance, the main aim of this study was to understand the impact of disturbances during autotrophic N removal under mainstream-like conditions at the level of transcription. In line with our findings on the process

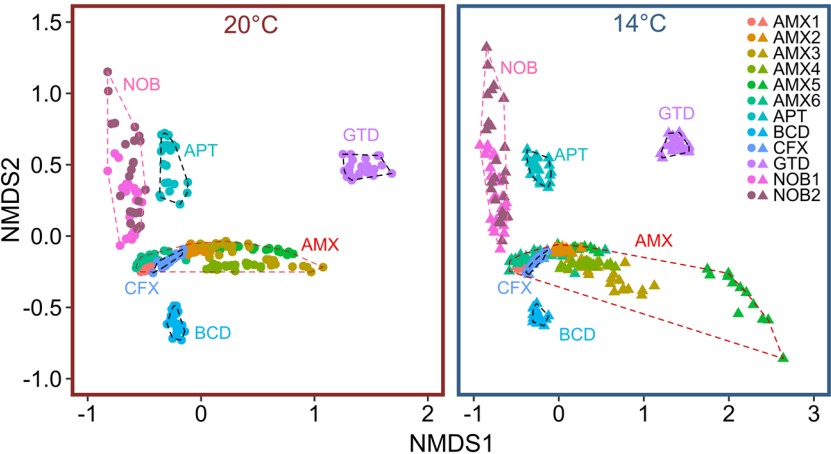

**Fig. 7 Transcriptomic differences on the MAG level.** nMDS (based on Jaccard dissimilarity) based on the COG expression profile for each MAG. Colors denote different MAGs and dashed lines (ordihull function) highlight genus cluster. Each dot represents the COG profile of the corresponding MAG on a respective time-point. Left panel presents the 20 °C and right panel the 14 °C experiment. Stress value: 0.074.

level, transcription was noticeably affected by disturbances at 20 °C, which is generally considered a more favorable temperature for anammox bacteria (Figs. 4 and 5). This might indicate a higher capacity of the community to actively react to stress at this temperature and a restrained physiology and stunted stress response at 14 °C. The observed, elevated return times to the referential state after exposure to 1.0 mg L$^{-1}$ h$^{-1}$ DO under 14 °C might also be an indication of temperature-restrained physiology.

Interestingly, while NH$_4^+$ removal rate was clearly affected by DO disturbances, they did not suppress the transcription of any of the key N cycle genes within the system, independent of temperature (Fig. 5B). Similar transcriptional responses patterns toward oxygen perturbations have been observed for anammox bacteria from coastal systems[46]. Together with our results on the process level, this suggests inhibition took place on the posttranscriptional level. Further evidence for this hypothesis is the rapidity of the functional response which was independent of the temperature same for reaching a maximum inhibition—if inhibition would have happened on the transcriptional level, the reduction in process rates presumably would have been less abrupt, as residual proteins would have continued to function until internal enzyme levels became depleted. Similarly, it would have presumably taken much longer for the system to resume its initial NH$_4^+$ removal efficiency if protein synthesis would first have to be initiated. Experimental studies on denitrifying bacteria in pure culture observed a timeframe of 10–24 h to establish a fully active denitrification enzyme system after a shift from aerobic to anaerobic conditions[47]. In our experiments, it took on average 37 min to return to referential state conditions (Supplementary Fig. 1A). Based on our observations, we speculate that during lower temperatures, not only the enzyme activity but also the enzyme inhibition is constrained which might have led to the less pronounced impact on the NH$_4^+$ removal rate during 14 °C. While the process response was thus not directly related to regulation of transcription, we nevertheless observed temperature-dependent differences in the transcription of key enzymes involved in the anammox process (Fig. 5B). We argue that the significantly higher expression of the *hdh* gene and lower expression of the *hzs* gene during the 14 °C experiment is probably an adaptation mechanism to lower temperatures. It is beyond dispute, that the conversion of hydrazine is the main energy yielding process in the anammox metabolism[40]. However, hydrazine is also toxic to anammox microorganisms if accumulated for long periods inside cells[48]. At colder temperatures, the metabolic machinery of anammox bacteria slows down, and this

may affect various steps of the process differently. To avoid hydrazine accumulation, the anammox cells appear to use transcriptional regulation to increase hydrazine dehydrogenase enzyme production and decrease hydrazine synthase enzyme production to ensure that all toxic hydrazine is converted efficiently to inert N$_2$. Reduced hydrazine synthase expression might have also contributed to the slow but steady performance loss during the 14 °C treatment.

Categorizing genes into their respective COG class revealed a clear upregulation of genes involved in the transcription, translation, replication, and posttranslational modification associated with the DO disturbance under the 20 °C regime (Fig. 5A). Especially, genes involved in bacterial stress tolerance displayed the biggest fluctuations in expression due to DO disturbances. The upregulation of heat-shock and cold-shock proteins allows for the maintenance of cellular processes during stress events and initiates a stress response cascade, which allows adaptation to harmful conditions[49,50]. These stress response systems did not react during the 14 °C experiment. Paradoxically, given the short-term disturbances in our experiment, this lack of response may have contributed to a higher resistance of the process observed at 14 °C. In this experiment, no long-term activity loss was observed after the short return phase, indicating that even at less favorable temperatures the anammox consortia suffered no permanent damage from the oxygen stress levels induced here. However, we speculate that with repeated or stronger disturbances, the lack of transcriptional response could result in reduced performance eventually.

We are aware that MAGs are computationally constructed entities and that caution should be taken in interpreting them as representative biological species. However, in order to investigate stress response of individual members from complex microbial communities, we consider this approach capable of offering new scientific insights that are also relevant for process engineers. It offers new opportunities to further our understanding how species-driven responses lead to process stability or failure in engineered ecosystems.

Using this unprecedented approach, we found that with only 19% relative abundance, anammox MAGs (AMX1-6; all *Candidatus* Brocadia) were nevertheless displaying highest transcript abundance in the system and drive the majority of the community metabolism, independent of the temperature regime (Fig. 6). Interestingly, this dominance in community transcript abundance was also previously reported for anammox bacteria in a laboratory-scale reactor, operated under sidestream conditions

(30 °C), but here they accounted for ~65% of the overall community[19,35]. Similar observations of disproportionally high transcriptional activity have also been made for marine AMX[51].

The presence of numerous *hao*-like genes and lack of *nir* genes in the retrieved anammox MAGs suggests that they pursue the recently proposed hydroxylamine-dependent anammox mechanism[52] (Supplementary Table 1). Here, $NO_2^-$ is reduced to hydroxylamine, which is then, together with $NH_4^+$, converted to hydrazine. The taxonomic classification of the MAGs as members of the *Candidatus Brocadia* spp. lineage supports this notion, as none of the previously studied representatives encode any known nitrite reductases[53]. Furthermore, we found that all anammox MAGs contain the nitric-oxide reductase gene (*nor*), which produces the homonymic enzyme, which reduces NO to $N_2O$ and potentially detoxifies the system[54]. We did not observe increased *nor* gene activity on the community level, which might be explained by the non-inhibiting but rather beneficial effect of NO on anammox bacteria[55].

The other retrieved MAGs together with the anammox MAGs shaped the backbone of a nearly closed N loop and could enhance overall N removal in the bioreactor[20] (Supplementary Table 1). Two-stage denitrification paired with anammox is a promising alternative to maximize biogas production[56]. However, it could also facilitate competition for denitrification-intermediates between anammox bacteria and other N key-players in the system[19].

Focussing on the stress response of individual MAGs revealed characteristic temporal COG transcription profiles for each MAG but also similarities between closest phylogenetic relatives (Fig. 7). Stress response on the transcriptional level appeared to be distinct even among species of the same genus but seems to be significantly different between phyla. While the two most dominant anammox MAGs displayed overall stable transcript abundance, transcriptional change was much larger during the 14 °C experiment for less abundant anammox MAGs. The biofilm carriers used in our experiment were exposed to mainstream ambient temperatures and seasonal temperature fluctuations for a period of 2 years. The most abundant species of the AMX consortia presumably have adaptations that allow them to thrive under these unstable conditions[57]. The less abundant AMX MAGs on the other hand are presumably stressed due to constant competition for available resources with the abundant AMX[58]. Thus, additional stress might have imposed the elevated stress responses of these MAGs.

Further studies of more severe stress and even actual system failures are needed to better understand the consequences of disturbance on diverse microbial community and transcriptional levels within this engineered ecosystem. Here, temperature was shown to change microbial community status in complex ways, but AMX biofilms proved resilient against short $O_2$ disturbances independent of temperature.

Long-term exposure to colder temperatures or prolonged DO disturbances might lead to completely different dynamics within the community but also on the individual level. We believe that our findings on transcriptional stress response advances our insight on the links between microbial community stress response, individual stress response, and process level failure. Furthermore, it emphasizes the value of molecular techniques paired with cutting-edge bioinformatics to understand individual stress response of key players in engineered ecosystems.

## Methods

**Reactor setup**. Triplicate 12 L temperature-controlled bioreactors for autotrophic N removal from mainstream WWTPs by biological anammox were operated in parallel. Reactors were inoculated with 1 kg of biofilm carriers (~$12 \times 12$ mm; surface area 1.200–1800 $m^2$ $m^{-3}$; FLUOPUR®, Wabag, Switzerland) with verified

anammox activity obtained from a pilot-scale (8 $m^3$) mainstream anammox reactor. The pilot-scale reactor is exposed to the seasonal temperature variations of the inflowing wastewater (~25–14 °C, average 17 °C). At the time of the experiment, it had operated successfully for over 2 years (~100 $g_{NH4-N}$ $m^3$ $d^{-1}$; ~5 mg $mg_{NH4-N}$ $L^{-1}$ h). A chemical oxygen demand-depleted effluent from another pilot scale (8 $m^3$) mainstream high rate activated sludge reactor (Eawag Dübendorf, Switzerland) was employed as influent[13]. The reactors were operated in SBR mode with online monitoring and data logging of sensor data on concentrations of $NH_4^+$, nitrate ($NO_3^-$), pH, temperature, conductivity, and redox potential[12,14,59]. Each SBR cycle of ~6.5 h was controlled by an automated control sequence that consisted of five steps: (1) settling (30 min), (2) effluent discharge, (3) feeding (6 L of pre-treated wastewater and addition of $NH_4Cl$ solution to a final concentration of 30 $mg_{NH4-N}$ $L^{-1}$), (4) reaction phase (variable duration, until $NH_4^+$ concentration reached 5 $mg_{NH4-N}$ $L^{-1}$), and (5) mixing (1 h to draw down residual $NO_2^-$). The volatile suspended solids concentration was ~1.9 gVSS/L. During the reaction phase, $NO_2^-$ was frequently added in small doses to maintain a limited concentration of 1–2 mg N $L^{-1}$ to allow continuous anammox activity and avoid substantial NOB growth[60]. Anoxic conditions were maintained by continuously purging with a mixture gas of 95% $N_2$ and 5% $CO_2$. With the constant addition of $NO_2^-$ and the continuous anoxic conditions, both pilot and lab-scale reactors simulated the AMX stage of a two-stage PN/A system.

**Pulse disturbance experiments under different temperature regimes**. Short-term DO perturbation experiments were conducted under different temperature regimes (20 °C, 14 °C) to investigate process variability and to assess temperature-dependent stress responses of anammox consortia (Fig. 1A). Temperatures were chosen based on the annual range of inflow temperatures of the pilot-scale reactor. For each temperature regime, the operational period of the reactors was 7 days. The temperature was controlled via a water heat jacket (Julabo GmbH, Germany).

After experiments at 20 °C were concluded, reactors were emptied, cleaned, and refilled with fresh carriers for the experiments at 14 °C to exclude any alterations of the biofilm due to prolonged exposure to the experimental reactor environment, changed temperature, or history of perturbation of the carriers. Before filling of the reactors, biofilm carrier samples were taken from the pilot scale reactor (Fig. 1: Inoculum 1/2). For both temperature regimes, reactors were first operated continuously without disturbance, i.e. baseline conditions, for 3 days to determine process stability and reproducibility in performance characteristics. A transient DO perturbation was applied by raising DO to 0.3 mg $L^{-1}$ for 1 h during the reaction phase of one SBR cycle. This was followed by 36 h of undisturbed operation to allow the system to fully recover. Reactors were then exposed to 1 h of 1.0 mg $L^{-1}$ of DO. Finally, reactors were operated for three additional SBR cycles of undisturbed operation. After both oxygen exposure periods, reactors were immediately flushed with additional $N_2$-gas to restore anoxic conditions.

Anammox activity was defined as the volumetric $NH_4^+$ removal rate during the reaction phases of the SBR cycles. The $NH_4^+$ removal rate at each time-point was calculated from the slope of a linear regression through the online concentration measurements by the $NH_4^+$ probes for a 10-min time interval. Under baseline operation, $NH_4^+$ removal rates (3.9 ± 0.7 $mg_{NH4-N}$ $L^{-1}$) were comparable to those of the pilot reactor (~5 mg $mg_{NH4-N}$ $L^{-1}$ h) from which the carriers were obtained. The pH varied between 7.4 and 8.0 during the SBR cycles over the course of the experiments. Temperature remained stable in both experiments (avg. standard deviation: 0.25 °C (20 °C); 0.32 °C (14 °C)).

Samples for offline analysis were taken 4–5 times during an SBR cycle. Here, $NH_4^+$ concentrations were also determined with photochemical test kits (Hach Lange GmbH, Düsseldorf, Germany, Test LCK303, spectrophotometer type LASA 26) to recalibrate online $NH_4^+$ sensors if necessary. $NO_2^-$ concentrations were determined using colorimetric test strips ($NO_2^-$-test, 0–10 $mg_{NO2-N}$ $L^{-1}$, Merck KGaA, Darmstadt, Germany).

**Biomass sampling, extraction, and sequencing**. A sample of five biofilm carriers per reactor was taken on specific time-points as shown in Fig. 1A, resulting in a total number of 56 samples. Samples were taken twice during undisturbed conditions (Baseline 1&2), 10 min and 50 min after the start of the DO disturbance (During stress 1&2) to capture the immediate and delayed transcriptional response and finally one in the subsequent SBR cycle (After stress) to assess the resilience of the system.

Biofilm carriers were immediately snap frozen in liquid nitrogen and stored at −80 °C for later DNA and RNA analysis.

Nucleic acids were extracted based on a method modified from Griffiths et al.[61]. Biofilm carriers ($n = 3$) were cut into small pieces in a liquid nitrogen bath. Carrier pieces were transferred to 1.5 mL Matrix E lysis tubes (MPbio) and 0.5 mL of both hexadecyltrimethylammonium bromide buffer and phenol:chloroform: isoamylalcohol (25:24:1, pH 6.8) were added. Cells were lysed in a FastPrep machine (MPbio), followed by nucleic acid precipitation with PEG 6000 on ice. Nucleic acids were washed three times with ethanol (70%) and dissolved in 50 μL of DEPC-treated RNAse-free water. Nucleic acids were separated overnight using a lithium-chloride (LiCl) solution (8 M). Resulting RNA pellets were purified and washed again three times with ethanol (70%) and dissolved in 50 μL of DEPC-treated RNAse-free water. DNA was precipitated via isopropanol from the LiCl supernatant. Pellets were washed two times with 70% ethanol and dissolved in

100 μL of DEPC-treated RNAse-free water. DNA quality was assessed by using agarose gel electrophoresis and a Nanodrop ND-2000c (Thermo Fisher Scientific, USA). Total RNA quality and quantity was subsequently checked using the Agilent TapeStation system (Agilent, Santa Clara, CA, USA) to ensure only high-quality nucleic acids were used for downstream analysis; 100 ng of RNA was used to construct strand-specific RNA-Seq libraries (Novogene, Hong Kong).

To track the transcriptional response of the microbial community, metatranscriptomic sequencing was performed on all 56 available RNA samples (Sample IDs Fig. 1) on the Illumina HiSeq 4000 platform to generate 150 bp paired-end reads. As the basis for genome assembly, and to assess the functional potential of the community, metagenomic sequencing was performed on nine DNA Samples (Baseline_2 at 20 °C and 14 °C pooled for each reactor and After_1.0 for both temperatures and each reactor). Metagenomes were sequenced on the Illumina NextSeq platform (Illumina, CA, USA) to generate 150 bp paired-end reads (350 bp mean insert size). All DNA and RNA sequencing was performed at Novogene, Hong Kong. Raw DNA and RNA sequences can be found on European Nucleotide Archive (ENA) under accession no. PRJEB36638.

**Metagenome assembly and annotation**. Raw DNA sequencing reads were quality controlled with FastQC[62] and Illumina adapters were trimmed, if necessary, with Trimmomatic[63]. Kaiju[64] was used to taxonomically assign the raw reads, using maximum exact matches of the query sequences translated to amino acids and protein database sequences. The reads were aligned against the NCBI non-redundant (NR) protein sequences from all bacteria, archaea, viruses, fungi, and microscopic-sized eukaryotes, respectively. Paired-end reads from each sample were assembled with Megahit[65] (~550000 contigs; ~N50: 1750 bp) and reads were mapped back via BBmap v35.92 (ref. [66]) with the parameters minid = 0.95 and ambig = random to assess assembly quality and coverage. Resulting bam files were handled and converted as needed using SAMtools1.3 (ref. [67]). Open reading frame (ORF) detection and subsequent gene annotation from assembled contigs was performed with prokka[68]. ORFs were queried against the SEED subsystems (pubseed.theseed.org; accessed July 2019), Clusters of Orthologous Groups (COG, https://www.ncbi.nlm.nih.gov/COG/; accessed June 2019), and Metacyc (https://metacyc.org/; accessed June 2019). To quantify the abundance of classified genes in the community, we mapped raw metagenomics reads back to the predicted and annotated ORFs of the assembled contigs. To express gene abundance, we divided the read counts by the length of each gene in kilobases. This led to reads per kilobase (RPK). We summed up all the RPK values in a sample and divided the number by 1,000,000. We divided the RPK values by the "per million" scaling factor leading to GPM. To ensure higher coverage for further transcriptomic mapping, we created an additional co-assembly of the nine metagenomic samples.

**Recovery and assessment of metagenome-assembled genomes (MAGs)**. High-quality trimmed reads from each sample were co-assembled into scaffolds using IDBA-UD[69] with the options --pre_correction --min_contig 1000. metaWRAP[70] binning and refinement modules were applied to the co-assembly. Completeness and contamination rates of the final bins were assessed using CheckM[37]. Bins were taxonomically classified using the metaWRAP classify module. Bin abundances were assessed using the metaWRAP quantification module. Here, raw reads were mapped against the putative genomes and abundance is expressed as the coverage of raw reads on the MAG. Phylogenetic analysis of the recovered draft genomes was performed with Phylosift v1.0.1, based on a set of 37 universal single-copy marker genes[38] using the 'phylosift -all' option. Thirty publicly available genomes closely related to the recovered draft genomes and ecosystem were downloaded to build a phylogenetic tree. Concatenated amino acid sequences of the marker genes were aligned with Phylosift, and a maximum likelihood phylogenetic tree was constructed with RAxML v8.2.4 with the PROTGAMMAAUTO model and 100 bootstraps[71]. MAGs can be accessed on European Nucleotide Archive (ENA) under accession no. PRJEB36638.

**Metatranscriptome analysis**. Analysis of metatranscriptomic reads was performed according to Lawson et al.[19]. Quality filtered paired-end reads were merged and rRNA sequences were filtered from the merged reads using SortMeRNA[72] v2.0, based on multiple bacterial, archaeal, and eukaryotic rRNA databases. Non-rRNA reads were mapped against the co-assembled metagenomic contigs (n = 9) using BBMap v35.92 with the parameters 'minid = 0.95', which specifies a minimum alignment identify of 95%. Ambiguous reads with multiple top-hit mapping locations were assigned to a random ORF ('ambig = random' option). Read counts were calculated for each predicted ORF using the FeatureCount option in the subread package[73]. Raw counts were normalized for sequencing depth and gene length and expressed as TPM[74] as a proxy for gene expression. The same workflow was applied to all recovered draft genomes to compare gene expression patterns across recovered MAGs. The relative transcript abundance (a) in TPM of the MAG was calculated according to Eq. (1) from the absolute number of mRNA reads that mapped to the MAG, divided by the genome length (bp) and normalized for sequencing depth (TPM):

$$a = \frac{\text{Average mapped cDNA (\#reads)}}{\text{MAG size}\left(\frac{\text{bp}}{1000}\right) \times \text{Sequencing depth}(\#\text{reads})/1000000}. \quad (1)$$

Furthermore, we investigated the relationship between MAG abundance and potential activity by calculating their ratio.

**Additional data analysis**. Normalized gene abundance and transcriptomic abundance tables from metagenome samples, draft genomes, and metatranscriptomes, respectively, were used for all further bioinformatics and statistical analysis. All data analyses, statistics, and visualization were conducted using R (R Core Team 2015, version 3.6)[75] with the packages vegan (version 2.5-6)[76] and ggplot2 (version 3.2.1)[77].

MAG-level transcription was compared using the functional transcript abundance at the level of COG classes with an nMDS analysis. Here, we summed up all transcript abundances (TPM) corresponding to a COG category for each MAG. The obtained COG abundance matrix was used as input for the metamds (with the Jaccard option) function of the vegan package.

**Statistics and reproducibility**. Data for the replicate reactors (n = 3) are given as averages and standard deviation of all values. Replicated reactors were operated under the exact same conditions over the whole period of the experiment and can be therefore taken as technical replicates. Comparisons between groups were determined with Student's t-test and Permanova. Linear regression models were used to assess significant changes in reactor operation.

## Data availability
Raw DNA and RNA sequences can be found on European Nucleotide Archive (ENA) under accession no. PRJEB36638.

All other data (Gene abundance tables as comma separated tables) is available in the Supplementary Data 1, 2 files, at the Eawag Research Data Institutional Collection (Eric) at https://doi.org/10.25678/0002MA[78] and upon reasonable request to the corresponding author (robert.niederdorfer@eawag.ch).

## Code availability
All R codes used in this study are available at https://doi.org/10.25678/0002MA[78] or upon request to the corresponding author (robert.niederdorfer@eawag.ch).

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

## Acknowledgements

This work was supported by funding from the Swiss national science foundation Synergia project ISOMOL: CRSII5_170876 to Moritz Lehmann (University of Basel). The authors would like to thank Moritz Lehmann (University of Basel) and Joachim Mohn (EMPA) for the helpful scientific discussions during the whole period of this study. Furthermore, we would like to acknowledge the Genetic Diversity Centre Zürich and the ETH Euler cluster team for their support and services.

## Author contributions

R.N., D.H., A.J., and H.B. designed the study. R.N. and D.H. performed the experiments. R.N. performed the sampling, sequencing, data analysis, and wrote the first draft of the manuscript. A.P. contributed the binning to this study. R.N. and H.B. wrote the manuscript with critical and helpful reviews from D.H., J.W., A.P., P.M., and B.S.

## Competing interests

The authors declare no competing interests.
