## [Peer Review File · Communications Biology]

Reviewers' comments:

Reviewer #1 (Remarks to the Author):

This manuscript investigated the response of microbial biofilms with verified anammox activity to oxygen shocks under favorable and cold temperature regimes. The topic is holding some potentials. But some of shortcomings have declined the consideration in its present for publication. Before acceptance some suggestion should be clarified by authors through revision.

Major comments:

(1) As we known, the dissolved oxygen and temperature were frequently discussed as the most critical factors with regards to stable operation of the anammox process. Why the authors still chose these two factors?

(2) The authors could cite some newest references in the MS

Specific comments:

Abstract :

(1) The important detailed data should be added in the abstract.

Introduction :

(1) Line 59-70: Recently, too many researches were focused on DO and temperature as the most critical factors to the anammox process, why did the authors study still?

(2) Line 71-74: The views expressed in this sentence are too absolute.

Materials and Methods :

(1) What is the VSS of the reactors?

(2) The biofilm carriers were taken from a mainstream anammox reactor, which was inoculated with ~25-17°C. But the authors set two temperature conditions during the entire experiment, i.e., 20 and 14°C. How did the authors control the temperatures? In the whole manuscript, the author did not mention the reason why they chose the two temperature conditions, which weaken the understanding of the significance of the study.

(3) Line 125-126: Why the authors cleaned and refilled with new biofilm carriers for the next experiments? What is the purpose?

(4) Line 125: We noticed that each experiment was operated for 7 days. As known, anammox bacteria is too sensitive to the environmental factors, so is the operating time too short of the reactors? We question the reliability of the data.

(5) Line 140-141: Why the authors set the cycle duration as 7.2 ± 2.0 h at 20 °C and 8.8 ± 2.3 h at 14°C?

(6) Line 150-151: How did the authors select the 56 samples? Did all the samples test for the metagenomics and metatranscriptomics analysis? The Fig. 1A was not clear enough, more details should be added in the MM.

Results and Discussion :

(1) Line 253-255: Please explain the reason of the NH_4^- remove rates was higher of R1 than others?

(2) Line 280-283: The microbial community composition and functional potential was stable, but why the reactor performance was changed? How to understand the "functional potential"?

(3) Phyla should be italicized.

(4) Line 325: Only 9 samples were taken to the metagenomic analysis?

(5) Line 344: Why the hzs could not be annotated in AMX4 and AMX5?

(6) Figure 5B: How did the authors set the data standardized?

(7) The discussion is let down by a lack of focus.

Reviewer #2 (Remarks to the Author):

This manuscript reports the response of anammox biofilms against dissolved oxygen and temperature disturbances. Genome-centric

metagenomics and metatranscriptomics were used to investigate the stress response on various biological levels. In general, this is a well written manuscript. My specific comments are below;

1. In line 41 of introduction, anammox under oxygen limited conditions- it implies that anammox could happen in the presence of small concentrations of oxygen which is not true. Please change to oxygen free conditions.
2. It is stated that temperature perturbations are common in mainstream anammox application. However, the temp fluctuations were done on anammox biofilms. Are anammox biofilms commonly used in mainstream applications?
3. Although this manuscript does not make it clear, it appears that the reactors simulated two stage anammox process. Which means, the reactors with anammox biofilm were primarily meant for anammox activity. Is this true and if yes, this does not reflect a true main stream environment because anammox community will coexist with zillions of other bacteria and the bacteria will face other perturbations such as organic carbon.
4. Hypothesis one that biofilms exposed to certain would experience longer effects could be rephrased to say that "Anammox biofilms"
5. The ammonium concentration was significantly higher in the reactor than the nitrite concentration. It appears that the reactor was operated under nitrite limited conditions. Why is that. Would that cause any stress on the community and community selection.
6. What is the basis of selecting the two temperature values. If we talk about mainstream, there are regions which experience mesophilic temperatures and there are geographical regions which experience temp lower than 14 degrees.
7. Even if the shock change was done in temp (7-day), I do understand the rationale behind this. For example with biofilm, the SRT is nearly infinite and hence, 7 days time period do not represent enough time to cause community shift, or recovery or consistent transcriptional changes.
8. Why there were differences in average SBR cycle length.
9. From lines 129 to 133, it appears that DO perturbations were only applied for 14 degree temp. This could be modified to reflect that DO perturbations were applied at both temperatures.
10. It appears that there is nearly 2-day time lag between when 0.3 DO was applied versus 1.0 mg/L DO was regime was applied. Are 2 days sufficient for the community to recover from the previous perturbation (e.g. 0.3 mg/L DO)? Additionally, would you call 0.3 mg/L as DO disturbance because many partial nitrification and anammox reactors are operated at 0.2 to 0.3 mg/L DO?
11. In line 193, it is stated that counts were first normalized for gene length and afterward for sequencing depth- how was it done?
12. Results: The whole discussion on DO perturbation at two temperatures seems routine. This type of findings have been published in the past and it is a proven fact that high DO inhibit anammox and the activity is reversible.
13. It is also surprising to not that DO completely inhibited NH₄ removal activity in all reactors at 20 °C (Figure 2A) but only partly at 14 °C (Figure 2B). Any explanation?
14. Interestingly, Nitrospira accounted for ~3% of total community even with such small doses of nitrite to the reactor- Any further explanation.
- 15 How complete were MAGs which were 10 % contaminated.
16. It was surprising to note that none of the anammox MAGs encoded for nirK or nirS genes. Why?
17. All the analysis involving COG pathways etc do not contribute much to the paper.
18. With all the analysis, so what, how this analyses helps in operational strategies?

Reviewer #3 (Remarks to the Author):

in their manuscript entitled "Temperature modulates stress response in mainstream anammox reactors" Niederdorfer and colleagues report metagenomic sequencing, transcriptomics and biochemical parameters of lab scale anammox reactors run at two temperatures (14°C & 20°C), and subjected to oxygen stress. The study is of interest, because knowledge of the microbial community response to (combinations of) different stressors allows for improved management of anammox reactors in the main line of wastewater treatment, where conditions are more variable than in controlled lab scale (or even pilot scale) studies.

Introduction:

The introduction is thorough and provides a clear justification for the study. I just have one minor suggestion for improvement. In line 74-80 the authors state that, in addition to metagenomics, metatranscriptomics can provide a clearer picture of community response in a time frame likely to short for community composition changes. In doing so previous work on anammox metagenomics is cited, but this doesn't entirely do justice to Lawson et al, who also employed transcriptomics in their study. There was a snapshot of a single condition, so it should be doable to highlight the merit of the authors' study design, without implicitly claiming to be the first to use metatranscriptomics in an anammox wastewater system.

Methods:

While the authors describe a number of different options for employing anammox reactors in wastewater treatment, it is unclear what type the pilot reactor where de biomass was sourced from is. The lack of AOB in the community suggests it is a two step system? In addition, it would be convenient for the reader to report the N removal activity in comparable units for the lab scale and pilot scale reactors. It looks like the N removal rate of the source reactor is comparable to the lab scale reactors, and I would state that in either line 108 or 252

line 174: please submit the raw reads to EBI as well, and associate them with bioproject number PRJEB36638. While MG-RAST is a valid repository for the data, the NCBI/EBI/DDBJ bioprject structure is designed so that all data belonging to a sequencing project can be easily retrieved by interested researchers.

Why did you choose to do gene annotation and abundance quantification on separate megahit assemblies, while doing MAG binning on an IDBA coassembly? If the assembly quality of the co-assembly is better, wouldn't that also provide better data for the binning independent analyses?

results:

Line 300-312: what does the gene abundance discussed in this paragraph mean? Is this number of metagenome reads mapped to annotated genes? or metatranscriptome reads?

Line 345-346: the physiological nitrite reductase in anammox is debated, and the inconsistent presence of nirS/nirK or neither throughout the anammox clades have led to the hypothesis that one of the hao-like enzymes conserved across all anammox genera could be the physiological nitrite reductase (<https://academic.oup.com/femsre/article/37/3/428/585200>)

Line 348-350: this is a surprising result, but may be by chance. The nxr of anammox bacteria does indeed cluster with the genes in Nitrospira and Nitrospina, but is not similar enough to them to be lost in assembly. it also is not more conserved than other anammox genes, so if the rest of the genomes could be assembled into distinct contigs, this should not be an issue for the nxr gene cluster. Are no

nxr genes identified in the metagenome?

The authors argue that the transcriptomic samples cluster by temperature (see note on figure 4 below), but it is unclear what is driving this clustering. Is a different subset of the community active, or are genes within the same organisms differentially regulated? A taxonomic classification of the transcriptomic reads might shed light on this.

(Metatranscriptomics has several tools to look beyond the broad classes of COG categories, such as describing log₂-fold change of expression of individual genes between treatments (possibly visualized using volcano plots if appropriate). Why did the authors decide not to show their data like this? Along this line: Does the label "prominent genes" in line 400 signify most changed or most expressed?

line 440-456: Do the NMDS plots in figure 7 really capture transcriptional regulation, or rather the inherent difference in COG categories between the taxonomic groups represented in the MAGs. In other words, if the plots were re-created using just the genome content, would they look similar?

Sup fig 8 and fig 7 seem to suggest that the expression patterns in AMX5 change substantially with temperature. Which COG classes (and which genes) drive the clear separation of the transcriptional response in AMX5?

Discussion:

line 529: the conversion of hydrazine to N₂ is discussed as a detox mechanism in this context, but it is also the main energy yielding process in anammox metabolism. The conversion to N₂ should always be optimized, because it is the payout for the investment of electrons hydrazine production requires.

Line 538-540: which fraction of the transcriptome reads could be mapped to the combined anammox organisms?

Stewart et al 2011 (<https://sfamjournals.onlinelibrary.wiley.com/doi/abs/10.1111/j.1462-2920.2010.02400.x>) showed that in the environmental setting of the ETSP oxygen minimum zone anammox bacteria (among others) are more transcriptionally active than would be expected based on their DNA abundance.

figures:

while visually appealing, the comparison of the relative abundance of the anammox species in figure 3B is challenging. It is also somewhat unclear how the taxonomic identification relates to the anammox MAGs. Given that supplementary figure 6 shows that all MAGs are *Brocadia* affiliated, does that imply that the detected *Jettenia* & *Kuenenia* sequences are unbinned, or that the reads are misclassified? I suggest changing the taxonomic identification of 8 different anammox species to the abundance of the 6 anammox MAGs, preferably with taxonomic designation

Figure 4 is somewhat strange to me, because it compares unrelated samples in each of the panels. As the reactors were cleaned and reinoculated between temperature treatments, comparing "reactor 3 20'C" to "reactor 3 14'C" is equally meaningful as comparing "reactor 3 20'C" to "reactor 1 14'C" and thus the figure gives a false temperature comparison. I'd rather expect that all data be plotted on a single NMDS, or that the figure consists of 2 panels, 1 for 20'C and 1 for 14'C.

In figure 5A, the "During 0.3/1" and "After 0.3/1" samples at 20'C look very similar, whereas the

"During 0.3/2" sample looks distinct. Is there an explanation for this variability?

Figure 5 shows the average TPM across the three reactors. How does the variability between reactors compare to the variability between time points?

Supplement

Supplemental figure 7 and the legend for supplemental figure 6 overlap

Sup Table 1: How do the authors distinguish between *hdh* and other hao-like genes in the anammox genomes? And when multiple genes are named (such as *nrfAH*) does that mean that both are present the indicated number of times?

Reviewers' comments:

Reviewer #1 (Remarks to the Author):

This manuscript investigated the response of microbial biofilms with verified anammox activity to oxygen shocks under favorable and cold temperature regimes. The topic is holding some potentials. But some of shortcomings have declined the consideration in its present for publication. Before acceptance some suggestion should be clarified by authors through revision.

Major comments:

(1) As we known, the dissolved oxygen and temperature were frequently discussed as the most critical factors with regards to stable operation of the anammox process. Why the authors still chose these two factors?

We choose these disturbances because, as also emphasized in the MS (L:69-83), they remain the most prominent factors responsible for process disruptions in full-scale AMX plants, and the issues are neither solved nor is the reason for the failures fully understood.

As the reviewer nicely pointed out, both disturbance factors were the focus of multiple scientific studies in the past years, however, the majority of these studies (T. Lotti, Kleerebezem, and van Loosdrecht 2015; T. Lotti et al. 2014; Cao et al. 2018; Joss et al. 2011; Lackner et al. 2015) were focusing on the effect of temperature and Oxygen on metrics of the anammox process performance. A few studies have looked also on how these disturbances affect community composition, but no previous study has investigated effects and mechanisms on the deeper levels of functional capacity and transcriptional activity (L:69+). Thus, despite the numerous excellent publications on these stressors, we saw a clear need for our study to contribute to a more systematic understanding how temperature and oxygen affect AMX bacteria. In addition, as our study represents a considerable increase in the methodological complexity of investigations, it was also important to already have a reasonable grasp of the expected response on the process level to make the investigation feasible.

(2) The authors could cite some newest references in the MS

We have updated or added a number of the most recent published articles on the topic.

Nsenga Kumwimba, M., Lotti, T., Şenel, E., Li, X. & Suanon, F. Anammox-based processes: How far have we come and what work remains? A review by bibliometric analysis. *Chemosphere* 238, 124627 (2020).

Wang, Y. et al. Exploring the effects of operational mode and microbial interactions on bacterial community assembly in a one-stage partial-nitrification anammox reactor using integrated multi-omics. *Microbiome* 7, 1–15 (2019).

Cao, S., Oehmen, A. & Zhou, Y. Denitrifiers in Mainstream Anammox Processes: Competitors or Supporters? *Environ. Sci. Technol.* 53, 11063–11065 (2019).

Zhang, L. & Okabe, S. Ecological niche differentiation among anammox bacteria. Water Res. 171, 115468 (2020).

Specific comments:

Abstract

(1) The important detailed data should be added in the abstract.

Thank you for this suggestion. We added relevant information in the abstract L:22-24 and L28-30, while also trying to maintain the maximum word count of 200 words.

Introduction

(1) Line 59-70: Recently, too many researches were focused on DO and temperature as the most critical factors to the anammox process, why did the authors study still?

As already stated in our reply to the major comment above, Temperature and Oxygen are the most prominent and crucial factors causing process disruptions in AMX reactors in practice. As the majority of previous studies was focusing on the process level or at best microbial community composition, we see a clear need to investigate how these two stressors affect the AMX consortia on the level of functional potential and transcriptional activity to obtain a deeper understanding what leads ultimately to the inhibition of the AMX process but also what induces the fast recovery, observed in all these studies.

We have revised the introduction (L 77-83) to state our motivation and objective more clearly.

“So far, the focus of these studies was to investigate the effect of applied oxygen or temperature disturbances on the process level and AMX efficiency, with a few studies providing further information on the microbial community composition (Laureni et al. 2016; Meng, Zhou, and Meng 2019; Cao et al. 2018). However, the stability problems have so far not been resolved, and a mechanistic understanding of the events that have led up to and potentially caused process failures has not yet emerged. This provides a motivation for studies using multi-omics approaches to understand what happens on the molecular level during performance failures and/or the ensuing recovery processes.”

(2) Line 71-74: The views expressed in this sentence are too absolute.

We agree with the reviewer and changed the "impact" of the sentence. L:77+

Materials and Methods

(1) What is the VSS of the reactors?

Reactor	Empty Carrier 10x [g]	Colonized Carrier 10x [g]	Rest after Volatilization 10x [g]	Dry Weight 10x [g]	VSS 10x [g]	VSS/Carrier [g]
R3	0.159	0.181	0.003	0.022	0.019	0.0019

R3	0.159	0.182	0.005	0.023	0.018	0.0018
R3	0.159	0.18	0.003	0.021	0.018	0.0018

- 1kg Carriers (wet weight) was put into 12L reactor Volume.
- Carrier wet weight = 0.0788 g/Carrier
- 1000g Carriers / 0.0788 g/Carrier = 12690 Carriers
- 12690 Carriers * 0.0018 VSS/Carrier = 22.84gVSS
- 22.84gVSS/12L reactor Volume = **1.9gVSS/L**

Thank you for pointing the lack of a VSS value in the MS, we added this information into the MM part (L:133-134). “The volatile suspended solids concentration was ~ 1.9 gVSS/L.”

(2) The biofilm carriers were taken from a mainstream anammox reactor, which was inoculated with ~25-17°C. But the authors set two temperature conditions during the entire experiment, i.e., 20 and 14°C. How did the authors control the temperatures? In the whole manuscript, the author did not mention the reason why they chose the two temperature conditions, which weaken the understanding of the significance of the study.

Thank you very much for this input. We have chosen these two temperature regimes as they reflect more favorable conditions (20°C) and less favorable conditions (14°C) in respect to the yearly average pilot-scale temperature of ~17°C. Furthermore, it roughly reflects the seasonal temperature maxima observed in the pilot-scale reactor as acknowledged in the MS (L: 143-146). The temperature was controlled via a water heat jacket.

“Temperatures were chosen based on the annual range of inflow temperatures of the pilot-scale reactor. For each temperature regime, the operational period of the reactors was 7 days. The temperature was controlled via a water heat jacket (Julabo GmbH, Germany).”

(3) Line 125-126: Why the authors cleaned and refilled with new biofilm carriers for the next experiments? What is the purpose?

We wanted to exclude any history of perturbation on the microbial community as a factor. Carriers from the 20°C were exposed to DO perturbations and could have developed an disturbance memory (T. Lotti et al. 2015), which could have compromised the outcome of the 14°C experiment. In the end, we wanted to ensure that both experiments have the exact same starting conditions. (L:148-150)

“After experiments at 20 °C were concluded, reactors were emptied, cleaned and refilled with fresh carriers for the experiments at 14 °C to exclude any alterations of the biofilm due to prolonged exposure to the experimental reactor environment, changed temperature or history of perturbation of the carriers.”

(4) Line 125: We noticed that each experiment was operated for 7 days. As known,

anammox bacteria is too sensitive to the environmental factors, so is the operating time too short of the reactors? We question the reliability of the data.

We were choosing this time-frames as we wanted to investigate the immediate transcriptional response on applied short term disturbances. As stated in the MS, we did not expect any community shift, given the long-term, stability of the AMX carriers. Studies have shown that transcriptional regulation could happen within minutes while also resilience can be observed within hours to days (Dalsgaard et al. 2014; Wells et al. 2017; Joss et al. 2011).

(5) Line 140-141: Why the authors set the cycle duration as 7.2h at 20 °C and 8.8h at 14 °C?

The length of the cycles was controlled by the NH₄⁺ removal rate and in the end on the AMX activity. Each of the cycles started with similar NH₄⁺ concentrations (30mg/L) and a new cycle was initiated only when the NH₄⁺ concentration reached a concentration below 5mg/l in the system.

(see L:132 “(4) reaction phase (variable duration, until NH₄⁺ concentration reached 5 mg_{NH4-N}·L⁻¹)”)

AMX bacteria have a temperature optimum of 25-30°C and display highest activity rates within this temperature window, the lower the temperature the lower the performance of AMX bacteria (T. Lotti, Kleerebezem, and van Loosdrecht 2015), which explains the effect on the length of the cycle in our experiments. We added information on this in the MS L: 291-293. “This explains the prolonged SBR cycles during the 14°C treatment, as cycle duration was directly dependent on the NH₄⁺ removal efficiency.”

(6) Line 150-151: How did the authors select the 56 samples? Did all the samples test for the metagenomics and metatranscriptomics analysis? The Fig. 1A was not clear enough, more details should be added in the MM.

Sampling times for metagenomics were selected to cover all experimental conditions, and all of these samples were sequenced. For metagenomics we only selected 3 Samples per reactor because we did not expect microbial community change in these experiments and had confirmed this with 16S amplicon sequencing data. We agree with the reviewer that the sampling criteria were not explicitly explained. We added a paragraph in the M&M (L:176-179). “Samples were taken twice during undisturbed conditions (Baseline 1&2), 10 minutes and 50 minutes after the start of the DO disturbance (During stress 1&2) to capture the immediate and delayed transcriptional response and finally one in the subsequent SBR cycle (After stress) to assess the resilience of the system.”

Results and Discussion

(1) Line 253-255: Please explain the reason of the NH₄⁺ remove rates was higher of R1 than others?

We are sorry but unfortunately, we cannot provide a precise answer to this question. All reactors were (i) fed with the same inflow, (ii) kept under the same temperature and

(iii) filled with exactly one kilogram of carrier material from the same source reactor. All reactors are identical builds and were operated under the exact same conditions to the best of our ability. However, like full scale waste water treatment plant reactors, these are biological systems interacting with complex, self-regulating technology, where small differences in conditions may lead to changes in performance. To quantify whether the over-performance of R1 or under-performance of R3 had an effect the SBR cycle variability and therefore influencing the SBR cycle time and comparability we calculated the coefficient of variation (CV) in NH_4^+ removal rate per cycle for each reactor and experiment. We did not observe significant differences in the per-cycle variation during baseline conditions between the reactors independent of the temperature ($p > 0.1$, two-sample t-test). In the end, we think that this observation underscores the importance of replication also in this kind of studies, which is still too often lacking.

(2) Line 280-283: The microbial community composition and functional potential was stable, but why the reactor performance was changed? How to understand the ???functional potential???

Functional potential refers to the functional profile derived from the metagenomic gene pool of the community, i.e. the genetic potential to perform certain metabolic functions present in the community. This functional potential remained stable over the course of the experiment. Since the presence of a gene does not imply its expression or a level of activity, we carefully use the term “potential”. This is also the reason why the reactor performance can change but not the functional potential or gene pool, since the actual activity is under several additional layers of control such as e.g. transcriptional and post-transcriptional regulation or inhibition. We have clarified this in L.312-314: “Our metagenome analysis confirmed that microbial community composition and functional potential (functional gene pool of the community) was stable”

(3) Phyla should be italicized.

We changed it accordingly.

(4) Line 325: Only 9 samples were taken to the metagenomic analysis?

Yes, as described in the Material and Methods we only sequenced 9 MG samples taken in different reactors and over the course of the experiment. As described above, since we expected community changes to play no role due to the short-term nature of the experiments and since the stability of the community was further confirmed by

amplicon sequencing, we focused on the transcriptome to follow the system response while the 9 metagenomes serve mostly to establish the functional potential of the community, show its stability and to provide the basis for genome reconstruction.

We clarified in M&M L:198+:

“To track the transcriptional response of the microbial community, metatranscriptomic sequencing was performed on all 56 available RNA samples (Sample IDs Figure 1) on the Illumina HiSeq 4000 platform to generate 150 bp paired-end reads. As a basis for genome assembly, and to assess the functional potential of the community, metagenomics sequencing was performed on nine DNA Samples (Baseline_2 at 20 °C and 14 °C pooled for each reactor and After_1.0 for both temperatures and each reactor).

(5) Line 344: Why the *hzs* could not be annotated in AMX4 and AMX5?"

In contrast to the other 4 AMX MAGs, none of the assembled contigs within the genome displayed a sufficient resemblance to published *hzs*. Specific reasons are not known, we assume that this is due to these regions randomly getting lost during the assembly and binning process to reconstruct the MAGs, as we are not aware of any specific difficulties associated with assembly of the *hzs* region. We manually blasted all contigs of these two MAGs to confirm the results obtained from the bioinformatics pipeline which confirmed the absence of *hzs* in these contigs (L: 376:378).

“However, *hzs* could not be annotated in AMX4 and AMX5, and may have been lost in the assembly. Manual BLAST analysis of the contigs of these MAGs confirmed the absence of *hzs* homologs.”

(6) Figure 5B: How did the authors set the data standardized?

We divided the read counts by the length of each gene in kilobases. This results in units of reads per kilobase (RPK). We summed up all up all the RPK values in a sample and divided by 1,000,000. This is the “per million” scaling factor. We divided the RPK values by the “per million” scaling factor leading to transcript per million (TPM). The same normalization was done for the metagenomic reads to obtain gene per million (GPM).

Here we subsetted the transcript table to genes that could be annotated as part of the N-cycle. Transcript per million is a widely used normalization technique, which allows to compare gene expression patterns of different samples or time-series like in our case. Therefore, and since the normalization is already explained in Methods, we have not made changes.

(7) The discussion is let down by a lack of focus.

We have endeavoured to improve the discussion in a number of ways: We introduced subheadings to the discussion to better lead the reader through the distinct sections of the discussion. In order to strengthen the focus, we further shortened the process level part of the discussion and put our focus on the transcriptional part and its relation to the findings on the process level.

We further reordered some sections to improve readability and built up more connection to the figures.

Reviewer #2 (Remarks to the Author):

This manuscript reports the response of anammox biofilms against dissolved oxygen and temperature disturbances. Genome-centric metagenomics and metatranscriptomics were used to investigate the stress response on various biological levels. In general, this is a well written manuscript. My specific comments are below;

1. In line 41 of introduction, anammox under oxygen limited conditions- it implies that anammox could happen in the presence of small concentrations of oxygen which is not true. Please change to oxygen free conditions.

We changed the sentence accordingly.

2. It is stated that temperature perturbations are common in mainstream anammox application. However, the temp fluctuations were done on anammox biofilms. Are anammox biofilms commonly used in mainstream applications?

As application of anammox to mainstream wastewater is still under development, the verdict on which solutions will ultimately prove successful is still open, but there are good arguments that biofilms are a promising approach. Autotrophic bacteria like nitrifiers and anammox bacteria are relative slow growing bacteria, which could lead to a biomass washout from the bioreactor and eventual process loss. However, these bacteria can form biofilms and this ability is useful for wastewater treatment. Here, engineers facilitate the formation of complex microbial communities and exploit the natural ability of AMX bacteria to form matrix enclosed biofilms on artificial carrier material or in absence of any substratum as suspended granular sludge, to extend biomass retention in engineered ecosystems (Strous et al. 1997; Flemming et al. 2016). In addition to the extended retention time of microbial biofilms, also lower space requirements, lower sludge production, enhanced ability to degrade recalcitrant compounds and the resilience to changes in the reactor configuration are only few of the benefits of the biofilm treatment systems (Wilderer and McSwain 2004; Chen and Chen 2000; Zhao et al. 2019). Several biofilm strategies exist for sidestream and mainstream anammox bioreactors, among them granules, trickling filters, rotating biological contactors and MBBRs (moving bed biofilm bioreactors). In this sense, all existing anammox systems indeed employ biofilms, although they may differ in important characteristics and we do not claim that our results can be extrapolated to other implementations.

3. Although this manuscript does not make it clear, it appears that the reactors simulated two stage anamox process. Which means, the reactors with anammox biofilm were primarily meant for anammox activity. Is this true and if yes, this does not reflect a true main stream environment because anammox community will coexist with zillions of other bacteria and the bacteria will face other perturbations such as organic carbon.

We modified the text to clarify that the bench-top experimental reactors resemble rather a two-stage system than a one-stage system (L: 137-139). “With the constant addition of NO₂⁻ and the continuous anoxic conditions, both pilot and lab-scale reactors simulated the AMX stage of a two-stage PN/A system.”

However, as our data shows, AMX are sharing their environment, as correctly acknowledged by the reviewer, with a huge diversity of other bacteria, which do not actively contribute to the AMX process itself but are involved in many other biochemical processes and also the complete N removal in the system. This is however the norm for many AMX systems as already shown by Lawson et al, Wang et al. and Speth et al. (Lawson et al. 2017; Speth et al. 2016; Wang et al. 2019).

Mainstream conditions can be classified with a temperature window between 10 and 25°C, an NH₄⁺ inflow concentration between 15-50 mgN·L⁻¹ and an NH₄⁺ effluent concentration of <1mgN·L⁻¹ (Hoekstra et al. 2019). As our reactors were fed with a chemical oxygen demand depleted inflow with very low NH₄⁺ and NO₂⁻ concentrations under temperatures below 20°C temperatures, it mimics in many facets the mainstream conditions in full scale WWTP.

4.Hypothesis one that biofilms exposed to certain would experience longer effects could be rephrased to say that "Anammox biofilms"

We changed the sentence accordingly.

5. The ammonium concentration was significantly higher in the reactor than the nitrite concentration. It appears that the reactor was operated under nitrite limited conditions. Why is that. Would that cause any stress on the community and community selection.

One of the main reasons to keep NO₂⁻ under limiting conditions is to suppress excessive growth of Nitrite oxidizing bacteria in the system, which could lead subsequently to competition for the available NO₂⁻ between AMX and NOB. Another reason to keep NO₂⁻ levels low is the inhibition potential of NO₂⁻ on the AMX process. Generally, Anammox is more vulnerable to nitrite inhibition than to ammonium inhibition (Isaka, Sumino, and Tsuneda 2007; Dapena-Mora et al. 2007). We added a statement in L:135-136. “During the reaction phase, NO₂⁻ was frequently added in small doses to maintain a limited concentration of 1-2 mg N L⁻¹ to allow continuous anammox activity and avoid substantial NOB growth.”

6. What is the basis of selecting the two temperature values. If we talk about mainstream, there are regions which experience mesophilic temperatures and there are geographical regions which experience temp lower than 14 degrees.

The reason we selected two different temperatures is the fact that under mainstream conditions the wastewater is strongly affected by the seasonal temperature changes. 14°C and 20°C cover the yearly temperature maxima the pilot-scale reactor is exposed to. There are many studies, which have shown a working AMX process under mesophilic conditions (Lackner et al. 2014). However, AMX also have been shown to grow on WWTP at low temperatures typical of moderate climates (10–15 °C) and with activities relevant for WWTP applications when nitrite is dosed (Laureni et al. 2015;

Tommaso Lotti et al. 2014). This has been clarified in the revised manuscript (L:143-144). “Temperatures were chosen based on the annual range of inflow temperatures of the pilot-scale reactor.”

7. Even if the shock change was done in temp (7-day), I do understand the rationale behind this. For example with biofilm, the SRT is nearly infinite and hence, 7 days time period do not represent enough time to cause community shift, or recovery or consistent transcriptional changes.

Indeed, community shift was not an expected outcome in this experiment, in which we focus on the transcriptional response instead. We chose the time-frame as we wanted to capture the immediate transcriptional response on applied short term disturbances which can happen in full scale plants randomly. As stated in the MS, we did not expect any community shift, given the long-term stability of the AMX carriers (L:311-316). Studies have shown that the system can be affected within minutes, changing from good performance to inhibition (Wells et al. 2017; Joss et al. 2011; Jin et al. 2012). Transcriptional response in bacteria is often observed on timescales of minutes to hours. Therefore we designed our experiment assuming that the transcriptional stress response would also be detectable within minutes (Dalsgaard et al. 2014).

8. Why there were differences in average SBR cycle length.

The length of the cycles was actively dependent on the NH₄⁺ removal rate and in the end on the AMX activity. Each of the cycles started with similar NH₄⁺ concentrations (30mg/L) and a new cycle was initiated only when the NH₄⁺ concentration reached a concentration below 5mg/l in the system. AMX bacteria have a temperature optimum of 25-30°C and display highest activity rates within this temperature window, the lower the temperature the lower the performance of AMX bacteria (T. Lotti, Kleerebezem, and van Loosdrecht 2015), which has an effect on the length of the cycle in our experiments. We have added some information on that in L:291-293 of the MS to make this more clear. “This explains the prolonged SBR cycles during the 14°C treatment, as cycle duration was directly dependent on the NH₄⁺ removal efficiency.”

9. From lines 129 to 133, it appears that DO perturbations were only applied for 14 degree temp. This could be modified to reflect that DO perturbations were applied at both temperatures.

We changed the sentence to clarify that DO perturbations were applied to both temperature regimes (L:151).

10. It appears that there is nearly 2-day time lag between when 0.3 DO was applied versus 1.0 mg/L DO was regime was applied. Are 2 days sufficient for the community to recover from the previous perturbation (e.g. 0.3 mg/L DO)? Additionally, would you call 0.3 mg/L as DO disturbance because many partial nitrification and anammox reactors are operated at 0.2 to 0.3 mg/L DO?

As we observed a strong resilience after the O₂ disturbances, even within the perturbed cycle we assumed 2 days time would be sufficient for the system to fully recover, even on the transcriptional level.

The pilot reactor in which the biofilm was grown, mimics the AMX stage of a two-stage AMX system and is therefore permanently anoxic, so for this system any oxygen pulse can be considered a disturbance. In Joss et al. 2011 (Joss et al. 2011) a full inhibition of the AMX process was observed at DO concentrations of 0.2mg/L. That is why we chose a DO level of 0.3mg for our system. 1.0mg was chosen as an extreme value of potential DO concentrations.

PNA one-stage reactors are flushed with Oxygen to allow sufficient work by AOB and conversion of enough NH₄⁺ to NO₂⁻. During the AMX activity phases, however, no oxygen is flushed into the system.

11. In line 193, it is stated that counts were first normalized for gene length and afterward for sequencing depth- how was it done?

We added more information on this topic to make the normalization of MG reads clearer. (L:224-228)

“To express gene abundance, we divided the read counts by the length of each gene in kilobases. This led to reads per kilobase (RPK). We summed up all up all the RPK values in a sample and divided the number by 1,000,000. We divided the RPK values by the “per million” scaling factor leading to Genes per million (GPM). To ensure higher coverage for further transcriptomic mapping, we created an additional co-assembly of the 9 metagenomic samples.”

12. Results: The whole discussion on DO perturbation at two temperatures seems routine. This type of findings have been published in the past and it is a proven fact that high DO inhibit anammox and the activity is reversible.

Indeed, these findings in themselves are not novel. However, in this study we focus on understanding how these responses come about. The novelty of our work lies in the information on the stress response and recovery on multiple biological levels within an engineered ecosystem. Nevertheless, the system response also has to be described on the previously studied process level. In our discussion, we always try to find a consensus between process, community and transcriptomic level. The revised and refocused discussion in the revised manuscript puts more emphasis on the novel aspects.

13. It is also surprising to not that DO completely inhibited NH₄ removal activity in all reactors at 20°C (Figure 2A) but only partly at 14°C (Figure 2B). Any explanation?

We kindly refer the reviewer to the discussion where we try to explain this phenomenon in more detail. We believe that, at 20°C the community has a higher capacity to actively react to stress (as the transcriptional response shows), while their response at 14°C is inhibited. When looking at the return times to a referential state, we can see longer recovery times during 14°C than during 20°C, which might be also an indication of temperature-restrained physiology. Furthermore, we believe that a

reduced hydrazine synthase expression might have also contributed to the slow but steady performance loss during the 14 °C treatment.

14. Interestingly, *Nitrospira* accounted for ~3% of total community even with such small doses of nitrite to the reactor- Any further explanation.

This is certainly a complex issue, but in our view the surprise was rather that *Nitrospira* reached such high in spite of the anaerobic conditions in the pilot scale reactor, while we would consider the nitrite supply quite sufficient for the observed *Nitrospira* abundance. There is simply not enough Oxygen for NOB to grow larger in numbers, but apparently enough oxygen enters through the reactor surface to allow microaerobic niches within the biofilm to sustain the 3% of NOB.

15 How complete were MAGs which were 10 % contaminated.

In terms of MAG completion and contamination, we followed the scientifically accepted guideline from Parks et al. (2015) (Parks et al. 2015). MAGs that are displaying a completeness of at least 50% with not more than 10% contamination are accepted as moderate complete. The higher the completeness the better the MAGs, contamination, however, has to remain always below 10% to ensure a good quality MAG. I would like to refer the reviewer to Table1, which displays the Completeness and Contamination levels of the MAGs used in this study. All MAGs are at least 80% completed and less than 6.6% contaminated making them moderate to substantially complete MAGs.

16. It was surprising to note that none of the anammox MAGs encoded for *nirK* or *nirS* genes. Why?

As highlighted already in the discussion (L:579-582) and now also in the result section (L:378-382) we believe the MAGs (all *Brocadia*) pursue the (known) *nirS/nirK* independent pathway, where hao-like enzymes are responsible for the first steps of the AMX process. Here, NO₂⁻ is reduced to hydroxylamine, which is then, together with NH₄⁺, converted to hydrazine (Oshiki et al. 2016). “Interestingly, except for AMX3, none of the anammox MAGs contained homologs of *nirK* or *nirS* genes, which are typically responsible for the first step of the anammox process (Kartal et al. 2013; Maalcke et al. 2016). The the inconsistent presence of *nirS/nirK* or neither throughout the anammox clades have led to the hypothesis that one of the hao-like enzymes conserved across all anammox genera could be the physiological nitrite reductase (Kartal et al. 2013).”

17. All the analysis involving COG pathways etc do not contribute much to the paper.

On this point, we disagree with the reviewer. By using the COG classification, we are able to summarily cover a large amount of important genes with designated function within the genomes and we can compare them on a similar level between all available genomes. Within the MS we investigate the changes in the COG classes over different levels. We find that, as expected, Energy production and conversion, Transcription and Translation are the COG classes with the largest gene pools. These, classes are the

most affected ones when it comes to temperature induced differences in transcript abundances. In the discussion, we try to find a consensus between these findings and the observations on the process level. Furthermore, we also see that individual MAGs have quite distinct temperature-induced differences in the transcript abundances of COG classes. Therefore, we believe that the COG analysis is well connected with other aspects of the study and contributes valuable information to the paper.

18. With all the analysis, so what, how this analyses helps in operational strategies?

Translating results on community molecular ecology to operational strategy is not a simple task, and we don't pretend to have solved the problems, but hope to have contributed to a better understanding that may inform further progress. For example, the noted differences in the expression of certain COG gene families between different temperatures and DO concentrations led to hypothesis on how AMX consortia deal in fundamentally different ways with stress at lower temperatures (Discussion), and we think this provides an important piece of information to understand process disturbance in systems undergoing large temperature fluctuations.

However, in the manuscript we also stress the fact, that this is one of the first studies that investigates the transcriptional response of AMX consortia to different stressors in combination (L:77-83; 94-98; 510-512). Further studies on multiple levels of stress response of AMX consortia and especially AMX enriched cultures could provide useful information on gene regulation and metabolite production. This in turn could allow us to develop early warning systems for reactors, which are based on stress molecules that are produced by the community in the face of a disturbance.

Reviewer #3 (Remarks to the Author):

in their manuscript entitled "Temperature modulates stress response in mainstream anammox reactors" Niederdorfer and colleagues report metagenomic sequencing, transcriptomics and biochemical parameters of lab scale anammox reactors run at two temperatures (14°C & 20°C), and subjected to oxygen stress. The study is of interest, because knowledge of the microbial community response to (combinations of) different stressors allows for improved management of anammox reactors in the main line of wastewater treatment, where conditions are more variable than in controlled lab scale (or even pilot scale) studies.

Introduction:

The introduction is thorough and provides a clear justification for the study. I just have one minor suggestion for improvement. In line 74-80 the authors state that, in addition to metagenomics, metatranscriptomics can provide a clearer picture of community response in a time frame likely to short for community composition changes. In doing so previous work on anammox metagenomics is cited, but this doesn't entirely do justice to Lawson et al, who also employed transcriptomics in their study. Theirs was a snapshot of a single condition, so it should be doable to highlight the merit of the authors' study design, without implicitly claiming to be the first to use metatranscriptomics in an anammox wastewater system.

We completely agree with the reviewer and now acknowledge the work by Lawson et al. (2017) but also Wang et al (Wang et al. 2019), who investigated the gene expression differences in different phases of the SBR cycle in a start-up phase of an PN/A reactor (L:87-94). We have defined the novelty of our approach more clearly in terms of applying these approaches to the study of community stress response in a replicated experimental design.

“Metatranscriptomic sequencing has already demonstrated its potential to provide detailed insights into the differences in gene expression of prokaryotic key players in AMX reactors under sidestream conditions(Lawson et al. 2017; Wang et al. 2019). Here we use, based on these pioneering studies, for the first time metatranscriptomics to investigate stress response in AMX reactors. In the framework of a fully replicated experimental design, we combine a multi-omics approach with a focus on metatranscriptomics with biochemical measurements.“

Methods:

While the authors describe a number of different options for employing anammox reactors in wastewater treatment, it is unclear what type the pilot reactor where de biomass was sourced from is. The lack of AOB in the community suggests it is a two step system?

As the reviewer correctly pointed out it is indeed a two-stage system. We updated the MS and added more clarification on this topic. (L:137-139) “With the constant addition of NO_2^- and the continuous anoxic conditions, both pilot and lab-scale reactors simulated the AMX stage of a two-stage PN/A system”.

In addition, it would be convenient for the reader to report the N removal activity in comparable units for the lab scale and pilot scale reactors. It looks like the N removal rate of the source reactor is comparable to the lab scale reactors, and I would state that in either line 108 or 252

Thank you very much for this input. We added the information on the activity of the pilot-scale reactor (L:124), and point out the equivalent performance of the lab-scale reactors (L:163).

Line 174: please submit the raw reads to EBI as well, and associate them with bioproject number PRJEB36638. While MG-RAST is a valid repository for the data, the NCBI/EBI/DDBJ bioprjject structure is designed so that all data belonging to a sequencing project can be easily retrieved by interested researchers.

We agree with the reviewer and submitted the raw data as well to the EBI server.

Why did you choose to do gene annotation and abundance quantification on separate megahit assemblies, while doing MAG binning on an IDBA coassembly? If the assembly quality of the co-assembly is better, wouldn't that also provide better data for the binning independent analyses?

We also performed a co-assembly of the 9MG samples as stated in L: 227-228 “To ensure higher coverage for further transcriptomic mapping, we created an additional co-assembly of the 9 metagenomic samples.” and 319-321 and used it to map MT reads against it.

Overall, IDBA led to the recovery of more high-quality bins in comparison to the MEGAHIT co-assembly. Therefore, we used two different co-assemblies.

Results:

Line 300-312: what does the gene abundance discussed in this paragraph mean? Is this number of metagenome reads mapped to annotated genes? or metatranscriptome reads?

Here we talk about the metagenomic reads that mapped to annotated genes. We clarified this now in the paragraph (L:335). “were the most abundant (summed metagenomic read abundances in GPM) (Supplementary Figure 3)”

Line 345-346: the physiological nitrite reductase in anammox is debated, and the inconsistent presence of nirS/nirK or neither throughout the anammox clades have led to the hypothesis that one of the hao-like enzymes conserved across all anammox genera could be the physiological nitrite reductase (<https://academic.oup.com/femsre/article/37/3/428/585200>)

Thank you very much for this input, we have integrated this information in L: 380+ “The the inconsistent presence of nirS/nirK or neither throughout the anammox clades have led to the hypothesis that one of the hao-like enzymes conserved across all anammox genera could be the physiological nitrite reductase”

Line 348-350: this is a surprising result, but may be by chance. The nxr of anammox bacteria does indeed cluster with the genes in Nitrospira and Nitrospina, but is not similar enough to them to be lost in assembly. it also is not more conserved than other anammox genes, so if the rest of the genomes could be assembled into distinct contigs, this should not be an issue for the nxr gene cluster. Are no nxr genes identified in the metagenome?

Thank you for these insights. We could indeed not detect nxr genes within the metagenomic assemblies. We used the MetaCyc database (metacyc.org) to manually curate all genes involved in the N-cycle to find the nxr genes, but unfortunately were not able to detect them. Based on your comment we have changed the phrasing here slightly.

The authors argue that the transcriptomic samples cluster by temperature (see note on figure 4 below), but it is unclear what is driving this clustering. Is a different subset of the community active, or are genes within the same organisms differentially regulated? A taxonomic classification of the transcriptomic reads might shed light on this.

(Metatranscriptomics has several tools to look beyond the broad classes of COG categories, such describing log₂-fold change of expression of individual genes between treatments (possibly visualized using volcano plots if appropriate). Why did the the

authors decide not to show their data like this?
Along this line: Does the label "prominent genes" in line 400 signify most changed or most expressed?

Thank you for this interesting question. We performed the Deseq2 approach to unravel differentially expressed genes. Unfortunately, this program was written mainly for experiments with a single species, under very controlled conditions and multiple replicates. The Deseq2 approach relies on the multifactorial analysis based on the differences between replicates of an untreated condition between treated conditions. Our experimental design was too complex for using this approach. We could only determine the differential expressed genes with regards to the factor temperature, as in this case we had enough statistical power to calculate them. Please have a look on the figure below.

Out of 756666 with nonzero total read count
adjusted p-value < 0.1
LFC > 0 (up) : 6002, 0.79%
LFC < 0 (down) : 13819, 1.8%
outliers [1] : 100, 0.013%
low counts [2] : 440055, 58%
(mean count < 2)

In total, we found 6000 genes that were upregulated and ~14000 genes that were downregulated. Even for temperature, analysis was only possible when taking the whole experiment into account, includes.g. ignoring the potential changes in gene expression due to the DO disturbances. Disentangling DO disturbances from the Temperature stress was not possible with the Deseq2 approach due to lack of sufficient numbers of baseline replicates (Baseline sample sequenced multiple times). This was the main reason to plot the differences in the normalized transcript abundance for each sample into an nmDS space, as it displays differences on the community transcript level in a very basic way and allows us to visually show one of our main results: **Temperature has an influence on the community transcriptional profile.**

Taxonomic classification the RNA reads were very consistent with the observations on the MG level. Majority of reads ($19.62414835 \pm 2.984692296$ %) could be assigned to the *Candidatus Brocadia Caroliensis* species, which is in line with the reads mapping to the AMX1 MAG but also the most abundant organism on the metagenome level. Overall, 42.2 ± 6.1 of the RNA reads could be assigned to a *Brocadia* genus, which is also agrees with the MG level (Supplementary Figure 2).

line 440-456: Do the NMDS plots in figure 7 really capture transcriptional regulation, or rather the inherent difference in COG categories between the taxonomic groups represented in the MAGs. In other words, if the plots were re-created using just the genome content, would they look similar?

Using the whole genome content to explore differences in the transcriptional regulation will lead to the problem that our genomes are of different sizes and harbor different numbers of genes. We would see a difference but we would not be able to compare them on a similar level. Using COG allowed us to narrow the gene pool down to large gene pool.

Certainly, the difference between gene transcription in the MAGs is to an extent determined by differences in the underlying genome. What we wanted to emphasize in this analysis, however was the similarity or dissimilarity of the transcriptional profile (through the rather coarse filter of COGs), and especially the difference in the temporal dynamics in different MAGs (which results only from transcriptional changes). The reason why we have chosen to take the COG classification is the fact, that we can with this approach cover a large range of genes by assigning them to designated functions and then compare transcriptional “investment” into these groups between all available genomes.

Sup fig 8 and fig 7 seem to suggest that the expression patterns in AMX5 change substantially with temperature. Which COG classes (and which genes) drive the clear separation of the transcriptional response in AMX5?

Overall, all COG classes display always a lower transcript abundance during the 14°C experiment than the 20°C in the AMX 5 MAG. All COG classes are driving to some extend the separation of the AMX5 MAG. However, COG class C, J and S were the most significant ones (Student’s t-test $p < 0.05$).

Discussion:

line 529: the conversion of hydrazine to N₂ is discussed as a detox mechanism in this context, but it is also the main energy yielding process in anammox metabolism. The conversion to N₂ should always be optimized, because it is the payout for the investment of electrons hydrazine production requires.

Thank you for pointing this out. We are aware that the N₂ production is the main energy yielding process, however, it is also true that Hydrazine is a toxic compound and producing too much from it, while not be able to process it could also lead to a toxification of the prokaryotic cell. We changed the MS (L:540-542) to point out the importance of this step for energy generation. “It is beyond dispute, that the conversion of Hydrazine is the main energy yielding process in the anammox metabolism (Kartal et al. 2013)”

Line 538-540: which fraction of the transcriptome reads could be mapped to the combined anammox organisms?

Stewart et al 2011

(<https://sfamjournals.onlinelibrary.wiley.com/doi/abs/10.1111/j.1462-2920.2010.02400.x>) showed that in the environmental setting of the ETSP oxygen minimum zone anammox bacteria (among others) are more transcriptionally active than would be expected based on their DNA abundance.

We had a close look on the MT reads that mapped on the AMX MAGs in comparison to the taxonomic classification of the raw reads. The majority of reads ($19.62414835 \pm 2.984692296$ %) could be assigned to the *Candidatus Brocadia Caroliensis* species, which is consistent with the reads mapping to the AMX1 MAG (~17%). Overall, 42.2 ± 6.1 of the RNA reads could be assigned to a *Brocadia* genus, which also agrees with the MG level (Supplementary Figure 2). All taxonomic classifications were thus in line with the findings on the MG level, therefore we are quite confident that there were no cross mappings of RNA reads on the AMX MAGs. We can indeed show that similar to the findings in Stewart et al., our AMX MAGs are more transcriptionally active than would be expected by their relative abundance in Figure 6. (L 437-439 : The two most abundant anammox MAGs (~18% relative abundance) together were significantly ($p < 0.05$, Student's t-test) responsible for the highest transcriptional activity during the 20 °C and 14 °C regime, respectively.)

We added the reference and a sentence to the MS (Ref 71 L:577-578: “Similar observations of disproportionately high transcriptional activity have also been made for marine AMX(Stewart, Ulloa, and Delong 2012).”

Figures:

while visually appealing, the comparison of the relative abundance of the anammox species in figure 3B is challenging. It is also somewhat unclear how the taxonomic identification relates to the anammox MAGs. Given that supplementary figure 6 shows that all MAGs are *Brocadia* affiliated, does that imply that the detected *Jettenia* & *Kuenenia* sequences are unbinned, or that the reads are misclassified? I suggest

changing the taxonomic identification of 8 different anammox species to the abundance of the 6 anammox MAGs, preferably with taxonomic designation

Thank you very much for this input. Indeed, we were not able to bin the *Kuenenia* and *Jettenia* species. However, removing them from Figure 3 would give an incorrect representation of the AMX community. Although, *Kuenenia* and *Jettenia* were not recovered as MAGs they are still part of the AMX consortia in the system. We added a statement in L 368:369 to make this point clear. “Unfortunately, we were not able to recover MAGs from the AMX genera *Candidatus Kuenenia* and *Candidatus Jettenia*.”

Figure 4 is somewhat strange to me, because it compares unrelated samples in each of the panels. As the reactors were cleaned and reinoculated between temperature treatments, comparing "reactor 3 20°C" to "reactor 3 14°C" is equally meaningful as comparing "reactor 3 20°C" to "reactor 1 14°C" and thus the figure gives a false temperature comparison. I would rather expect that all data be plotted on a single NMDS, or that the figure consists of 2 panels, 1 for 20°C and 1 for 14°C.

Thank you for this suggestion but we disagree on this statement. Cleaning and reinoculating was indeed the best strategy to ensure the greatest similarity in starting conditions, because if this had not been done, the experiments on the other temperature would have been performed on a community that had already been in the experimental reactors longer, at a different temperature, and experienced oxygen stress episodes. Our intention was to show that temperature has an influence on the community-wide transcription independent of the reactors. Given the differences between reactors that are also seen in Figure 1, separating the by reactors provides a clearer view. Similar to our findings on the process level we want to show with this figure both the general temperature effect and that the impact of DO has a stronger influence during the 20°C experiment. Both findings are further supported by multiple lines of analysis in the MS and in other figures.

In figure 5A, the "During 0.3/1" and "After 0.3/1" samples at 20°C look very similar, whereas the "During 0.3/2" sample looks distinct. Is there an explanation for this variability?

We can only speculate, but believe that the system adapted here to the mild DO perturbation. Switching it off led again to a “shock” which could lead to observed differences. When, looking on the 1mg DO, we can see that the system is constantly upregulated and no adaptation is happening. As soon as the DO stress is over we return to a “normal” expression pattern. It would require additional experimentation to provide a more confident answer to this question.

Figure 5 shows the average TPM across the three reactors. How does the variability between reactors compare to the variability between time points?

The left plot displays the sum of all TPM from genes of the COG category K, which also displayed the highest fluctuations during the experiment. The right plot displays the cumulative TPM for the Hydrazine dehydrogenase between the three Reactors over the whole experiment. We did not find significant differences between the reactors for any COG category, nor for N-genes.

Supplement

Supplemental figure 7 and the legend for supplemental figure 6 overlap
We changed it.

Sup Table 1: How do the authors distinguish between *hdh* and other hao-like genes in the anammox genomes? And when multiple genes are named (such as *nrfAH*) does that mean that both are present the indicated number of times?

Here we counted the copies of annotated genes present in the genome. Prokka annotated the Hydrazine dehydrogenase and Hydroxylamine Oxidoreductase and we manually blasted the sequences to confirm the annotation.

It actually means that at least one copy is present. We changed the Supplementary Table 1 to make this point clear. Thank you for pointing that out.

- Cao, Yeshi, Bee Hong Kwok, Mark C.M. Van Loosdrecht, Glen Daigger, Hui Yi Png, Wah Yuen Long, and Ooi Kian Eng. 2018. "The Influence of Dissolved Oxygen on Partial Nitrification/ Anammox Performance and Microbial Community of the 200,000 M3/d Activated Sludge Process at the Changi Water Reclamation Plant (2011 to 2016)." *Water Science and Technology* 78 (3): 634–43. <https://doi.org/10.2166/wst.2018.333>.
- Chen, C. Y., and S. D. Chen. 2000. "Biofilm Characteristics in Biological Denitrification Biofilm Reactors." *Water Science and Technology* 41 (4–5): 147–54. <https://doi.org/10.2166/wst.2000.0438>.
- Dalsgaard, Tage, Frank J. Stewart, Bo Thamdrup, Loreto De Brabandere, Niels Peter Revsbech, Osvaldo Ulloa, Don E. Canfield, and Edward F. Delong. 2014. "Oxygen at Nanomolar Levels Reversibly Suppresses Process Rates and Gene Expression in Anammox and Denitrification in the Oxygen Minimum Zone off Northern Chile." Edited by Douglas G. Capone. *MBio* 5 (6): 1–14. <https://doi.org/10.1128/mBio.01966-14>.
- Dapena-Mora, A., I. Fernández, J. L. Campos, A. Mosquera-Corral, R. Méndez, and M. S.M. Jetten. 2007. "Evaluation of Activity and Inhibition Effects on Anammox Process by Batch Tests Based on the Nitrogen Gas Production." *Enzyme and Microbial Technology* 40 (4): 859–65. <https://doi.org/10.1016/j.enzmictec.2006.06.018>.
- Flemming, Hans Curt, Jost Wingender, Ulrich Szewzyk, Peter Steinberg, Scott A. Rice, and Staffan Kjelleberg. 2016. "Biofilms: An Emergent Form of Bacterial Life." *Nature Reviews Microbiology* 14 (9): 563–75. <https://doi.org/10.1038/nrmicro.2016.94>.
- Hoekstra, Maaïke, Stefan P. Geilvoet, Tim L.G. Hendrickx, Charlotte S. van Erp Taalman Kip, Robbert Kleerebezem, and Mark C.M. van Loosdrecht. 2019. "Towards Mainstream Anammox: Lessons Learned from Pilot-Scale Research at WWTP Dokhaven." *Environmental Technology (United Kingdom)* 40 (13): 1721–33. <https://doi.org/10.1080/09593330.2018.1470204>.
- Isaka, Kazuichi, Tatsuo Sumino, and Satoshi Tsuneda. 2007. "High Nitrogen Removal Performance at Moderately Low Temperature Utilizing Anaerobic Ammonium Oxidation Reactions." *Journal of Bioscience and Bioengineering*. <https://doi.org/10.1263/jbb.103.486>.
- Jin, Ren Cun, Guang Feng Yang, Jin Jin Yu, and Ping Zheng. 2012. "The Inhibition of the Anammox Process: A Review." *Chemical Engineering Journal* 197: 67–79. <https://doi.org/10.1016/j.cej.2012.05.014>.
- Joss, Adriano, Nicolas Derlon, Clementine Cyprien, Sabine Burger, Ilona Szivak, Jacqueline Traber, Hansruedi Siegrist, and Eberhard Morgenroth. 2011. "Combined Nitrification-Anammox: Advances in Understanding Process Stability." *Environmental Science and Technology* 45 (22): 9735–42. <https://doi.org/10.1021/es202181v>.
- Kartal, Boran, Naomi M. De Almeida, Wouter J. Maalcke, Huub J.M. Op den Camp, Mike S.M. Jetten, and Jan T. Keltjens. 2013. "How to Make a Living from Anaerobic Ammonium Oxidation." *FEMS Microbiology Reviews* 37 (3): 428–61. <https://doi.org/10.1111/1574-6976.12014>.
- Lackner, Susanne, Eva M. Gilbert, Siegfried E. Vlaeminck, Adriano Joss, Harald Horn, and Mark C.M. van Loosdrecht. 2014. "Full-Scale Partial Nitrification/Anammox Experiences - An Application Survey." *Water Research* 55 (0): 292–303. <https://doi.org/10.1016/j.watres.2014.02.032>.
- Lackner, Susanne, Samuel Welker, Eva M. Gilbert, and Harald Horn. 2015. "Influence

- of Seasonal Temperature Fluctuations on Two Different Partial Nitritation-Anammox Reactors Treating Mainstream Municipal Wastewater.” *Water Science and Technology* 72 (8): 1358–63. <https://doi.org/10.2166/wst.2015.301>.
- Laureni, Michele, Per Falås, Orlane Robin, Arne Wick, David G. Weissbrodt, Jeppe Lund Nielsen, Thomas A. Ternes, Eberhard Morgenroth, and Adriano Joss. 2016. “Mainstream Partial Nitritation and Anammox: Long-Term Process Stability and Effluent Quality at Low Temperatures.” *Water Research* 101: 628–39. <https://doi.org/10.1016/j.watres.2016.05.005>.
- Laureni, Michele, David G. Weissbrodt, Ilona Szivák, Orlane Robin, Jeppe Lund Nielsen, Eberhard Morgenroth, and Adriano Joss. 2015. “Activity and Growth of Anammox Biomass on Aerobically Pre-Treated Municipal Wastewater.” *Water Research* 80: 325–36. <https://doi.org/10.1016/j.watres.2015.04.026>.
- Lawson, Christopher E., Sha Wu, Ananda S. Bhattacharjee, Joshua J. Hamilton, Katherine D. McMahon, Ramesh Goel, and Daniel R. Noguera. 2017. “Metabolic Network Analysis Reveals Microbial Community Interactions in Anammox Granules.” *Nature Communications* 8 (May): 1–12. <https://doi.org/10.1038/ncomms15416>.
- Lotti, T., R. Kleerebezem, J. M. Abelleira-Pereira, B. Abbas, and M. C.M. van Loosdrecht. 2015. “Faster through Training: The Anammox Case.” *Water Research* 81: 261–68. <https://doi.org/10.1016/j.watres.2015.06.001>.
- Lotti, T., R. Kleerebezem, Z. Hu, B. Kartal, M. S.M. Jetten, and M. C.M. van Loosdrecht. 2014. “Simultaneous Partial Nitritation and Anammox at Low Temperature with Granular Sludge.” *Water Research* 66: 111–21. <https://doi.org/10.1016/j.watres.2014.07.047>.
- Lotti, T., R. Kleerebezem, and M. C.M. van Loosdrecht. 2015. “Effect of Temperature Change on Anammox Activity.” *Biotechnology and Bioengineering* 112 (1): 98–103. <https://doi.org/10.1002/bit.25333>.
- Lotti, Tommaso, Robbert Kleerebezem, Charlotte Van Erp Taalman Kip, Tim L.G. Hendrickx, Jans Kruit, Maaïke Hoekstra, and Mark C.M. Van Loosdrecht. 2014. “Anammox Growth on Pretreated Municipal Wastewater.” *Environmental Science and Technology* 48 (14): 7874–80. <https://doi.org/10.1021/es500632k>.
- Maalcke, Wouter J., Joachim Reimann, Simon De Vries, Julea N. Butt, Andreas Dietl, Nardy Kip, Ulrike Mersdorf, et al. 2016. “Characterization of Anammox Hydrazine Dehydrogenase, a Key -Producing Enzyme in the Global Nitrogen Cycle.” *Journal of Biological Chemistry* 291 (33): 17077–92. <https://doi.org/10.1074/jbc.M116.735530>.
- Meng, Yabing, Zhongbo Zhou, and Fangang Meng. 2019. “Impacts of Diel Temperature Variations on Nitrogen Removal and Metacommunity of Anammox Biofilm Reactors.” *Water Research* 160: 1–9. <https://doi.org/10.1016/j.watres.2019.05.021>.
- Oshiki, Mamoru, Muhammad Ali, Kaori Shinyako-Hata, Hisashi Satoh, and Satoshi Okabe. 2016. “Hydroxylamine-Dependent Anaerobic Ammonium Oxidation (Anammox) by ‘Candidatus Brocadia Sinica.’” *Environmental Microbiology* 18 (9): 3133–43. <https://doi.org/10.1111/1462-2920.13355>.
- Parks, Donovan H., Michael Imelfort, Connor T. Skennerton, Philip Hugenholtz, and Gene W. Tyson. 2015. “CheckM: Assessing the Quality of Microbial Genomes Recovered from Isolates, Single Cells, and Metagenomes.” *Genome Research* 25 (7): 1043–55. <https://doi.org/10.1101/gr.186072.114>.
- Speth, Daan R., Michiel H. In’T Zandt, Simon Guerrero-Cruz, Bas E. Dutilh, and Mike S.M. Jetten. 2016. “Genome-Based Microbial Ecology of Anammox Granules in

- a Full-Scale Wastewater Treatment System.” *Nature Communications* 7. <https://doi.org/10.1038/ncomms11172>.
- Stewart, Frank J., Osvaldo Ulloa, and Edward F. Delong. 2012. “Microbial Metatranscriptomics in a Permanent Marine Oxygen Minimum Zone.” *Environmental Microbiology*. <https://doi.org/10.1111/j.1462-2920.2010.02400.x>.
- Strous, Marc, Eric Van Gerven, J. Gijs Kuenen, and Mike Jetten. 1997. “Effects of Aerobic and Microaerobic Conditions on Anaerobic Ammonium- Oxidizing (Anammox) Sludge.” *Applied and Environmental Microbiology* 63 (6): 2446–48.
- Wang, Yulin, Qigui Niu, Xu Zhang, Lei Liu, Yubo Wang, Yiqiang Chen, Mishty Negi, Daniel Figeys, Yu You Li, and Tong Zhang. 2019. “Exploring the Effects of Operational Mode and Microbial Interactions on Bacterial Community Assembly in a One-Stage Partial-Nitritation Anammox Reactor Using Integrated Multi-Omics.” *Microbiome* 7 (1): 1–15. <https://doi.org/10.1186/s40168-019-0730-6>.
- Wells, G. F., Y. Shi, M. Laurenzi, A. Rosenthal, I. Szivák, D. G. Weissbrodt, A. Joss, H. Buergermann, D. R. Johnson, and E. Morgenroth. 2017. “Comparing the Resistance, Resilience, and Stability of Replicate Moving Bed Biofilm and Suspended Growth Combined Nitritation-Anammox Reactors.” *Environmental Science and Technology* 51 (9): 5108–17. <https://doi.org/10.1021/acs.est.6b05878>.
- Wilderer, P. A., and B. S. McSwain. 2004. “The SBR and Its Biofilm Application Potentials.” *Water Science and Technology* 50 (10): 1–10. <https://doi.org/10.2166/wst.2004.0596>.
- Zhao, Yunpeng, Ying Feng, Liming Chen, Zhao Niu, and Sitong Liu. 2019. “Genome-Centered Omics Insight into the Competition and Niche Differentiation of *Ca. Jettenia* and *Ca. Brocadia* Affiliated to Anammox Bacteria.” *Applied Microbiology and Biotechnology* 103 (19): 8191–8202. <https://doi.org/10.1007/s00253-019-10040-9>.

Reviewers' comments:

Reviewer #1 (Remarks to the Author):

My comments have been well addressed. It can be accepted for publication.

Reviewer #2 (Remarks to the Author):

Congratulations on the excellent revision. All comments have been nicely addressed.

Reviewer #3 (Remarks to the Author):

I thank the authors for their thorough responses to my earlier feedback. I am largely satisfied with the changes to the manuscript, but there are two points where I disagree with the authors explanation to the extent that it warrants further discussion. They concern data as displayed in figure 3 and 4. For easy referral, I have included both my original comment, the author's response, and my additional comment for either point.

Figure 3:

Round 1 review

While visually appealing, the comparison of the relative abundance of the anammox species in figure 3B is challenging. It is also somewhat unclear how the taxonomic identification relates to the anammox MAGs. Given that supplementary figure 6 shows that all MAGs are *Brocadia* affiliated, does that imply that the detected *Jettenia* & *Kuenenia* sequences are unbinned, or that the reads are misclassified? I suggest changing the taxonomic identification of 8 different anammox species to the abundance of the 6 anammox MAGs, preferably with taxonomic designation

Author response

Thank you very much for this input. Indeed, we were not able to bin the *Kuenenia* and *Jettenia* species. However, removing them from Figure 3 would give an incorrect representation of the AMX community. Although, *Kuenenia* and *Jettenia* were not recovered as MAGs they are still part of the AMX consortia in the system. We added a statement in L 368:369 to make this point clear. "Unfortunately, we were not able to recover MAGs from the AMX genera *Candidatus Kuenenia* and *Candidatus Jettenia*."

Round 2 review

I might be misunderstanding the data here, but I think *Jettenia* and *Kuenenia* might not be present in the system. If I understand the analysis correctly, figure 3 is based on read assignment using Kaiju, which can only identify previously known species. If the *Brocadia* organisms in this reactor happen to have gene content that is not present in the previously published *Brocadia* sp., but happens to be found in either *Jettenia* or *Kuenenia* sp. those organisms will be "detected". The authors could test this by mapping the reads identified as either *Jettenia* or *Kuenenia* on the retrieved MAGs, and see how well they match (i.e. whether these MAGs are the actual origin of the reads). Taxonomic assignment of unbinned read is challenging and error prone, and if looking at your data from two different angles gives you different answers, I think it's worth investigating whether both angles are equally reliable.

Figure 4:

Round 1 review

Figure 4 is somewhat strange to me, because it compares unrelated samples in each of the panels. As the reactors were cleaned and reinoculated between temperature treatments, comparing "reactor 3 20°C" to "reactor 3 14°C" is equally meaningful as comparing "reactor 3 20°C" to "reactor 1 14°C" and thus the figure gives a false temperature comparison. I would rather expect that all data be plotted on a single NMDS, or that the figure consists of 2 panels, 1 for 20°C and 1 for 14°C.

Author Response

Thank you for this suggestion but we disagree on this statement. Cleaning and reinoculating was indeed the best strategy to ensure the greatest similarity in starting conditions, because if this had not been done, the experiments on the other temperature would have been performed on a community that had already been in the experimental reactors longer, at a different temperature, and experienced oxygen stress episodes. Our intention was to show that temperature has an influence on the community-wide transcription independent of the reactors. Given the differences between reactors that are also seen in Figure 1, separating the by reactors provides a clearer view. Similar to our findings on the process level we want to show with this figure both the general temperature effect and that the impact of DO has a stronger influence during the 20°C experiment. Both findings are further supported by multiple lines of analysis in the MS and in other figures.

Round 2 review

I'm not sure this is a point where we can agree to disagree, as I am a little worried that this could be unintentional data manipulation. If your effect goes away when including all replicates, it might not be real. I can not think of a type of experiment where replicates are done, and instead of comparing all replicates of one condition with all replicates of the other condition, compare them experiment by experiment (which is what you're doing here). As you have cleaned the reactors in between, conditions the replicates should be independent (i.e. "by reactor" should be meaningless) and you have no justification for comparing them as you do, instead of any other permutation.

I agree that the spread of the points for 14°C experiments is smaller than for the 20°C experiments, indicating a smaller response to the DO perturbation, and expect that this effect is also clearly visible in an NMDS including all samples. Where I disagree is that you can use this visualization to show that the clustering of the points is driven by temperature (rather than by inoculum, for example). A NMDS including all data could clarify this.

I'm worried about this for two reasons:

- 1) if all data points group by experiment, your inoculum effect is larger than your temperature effect, and the figure (and data interpretation) should reflect that.
- 2) if the data clearly group by reactor, there is a batch effect as a result of the systems not being completely identical, and you might need to correct for that.

Reviewers' comments:

Reviewer #1 (Remarks to the Author):

My comments have been well addressed. It can be accepted for publication. Thank you for the positive assessment.

Reviewer #2 (Remarks to the Author):

Congratulations on the excellent revision. All comments have been nicely addressed. Thank you for the positive assessment.

Reviewer #3 (Remarks to the Author):

I thank the authors for their thorough responses to my earlier feedback. I am largely satisfied with the changes to the manuscript, but there are two points where I disagree with the authors explanation to the extent that it warrants further discussion. They concern data as displayed in figure 3 and 4. For easy referral, I have included both my original comment, the author's response, and my additional comment for either point.

Figure 3:

Round 1 review

While visually appealing, the comparison of the relative abundance of the anammox species in figure 3B is challenging. It is also somewhat unclear how the taxonomic identification relates to the anammox MAGs. Given that supplementary figure 6 shows that all MAGs are Brocadia affiliated, does that imply that the detected Jettenia & Kuenenia sequences are unbinned, or that the reads are misclassified? I suggest changing the taxonomic identification of 8 different anammox species to the abundance of the 6 anammox MAGs, preferably with taxonomic designation

Author response

Thank you very much for this input. Indeed, we were not able to bin the Kuenenia and Jettenia species. However, removing them from Figure 3 would give an incorrect representation of the AMX community. Although, Kuenenia and Jettenia were not recovered as MAGs they are still part of the AMX consortia in the system. We added a statement in L 368:369 to make this point clear. "Unfortunately, we were not able to recover MAGs from the AMX genera Candidatus Kuenenia and Candidatus Jettenia."

Round 2 review

I might be misunderstanding the data here, but I think Jettenia and Kuenenia might not be present in the system. If I understand the analysis correctly, figure 3 is based on read assignment using Kaiju, which can only identify previously known species. If the Brocadia organisms in this reactor happen to have gene content that is not present in the previously published Brocadia sp., but happens to be found in either Jettenia or Kuenenia sp. those organisms will be "detected". The authors could test this by mapping the reads identified as either Jettenia or Kuenenia on the retrieved MAGs, and see how well they match (i.e. whether these MAGs are the actual origin of the

reads). Taxonomic assignment of unbinned read is challenging and error prone, and if looking at your data from two different angles gives you different answers, I think it's worth investigating whether both angles are equally reliable.

We thank the reviewer for bringing potential issues with the read based taxonomic assignment to our attention. We completely agree with the reviewer's opinion that taxonomic assignment on the read level, especially down to species level, can be error prone and lead to misinterpretation of the data, but had originally decided to report the outcome of the analysis as-is. The assignment of reads to *Kuenenia* and *Jettenia* was consistently obtained with different tools (Kaiju, Kraken).

As suggested by the reviewer, we mapped the extracted *Kuenenia* and *Jettenia* raw reads to our *Brocadia* MAGs and also to the available draft genomes of *Kuenenia* and *Jettenia* (PRJEB22746 and PRJNA534024, respectively). We obtained only a low proportion of assignment of these reads to the *Brocadia* bins, the highest percentage of matches were obtained for AMX1, where 3% (127 reads) of *Kuenenia* raw reads and 2% (58 reads) *Jettenia* reads mapped and for AMX 3 with ~4 % *Kuenenia* and 4% *Jettenia* mapped to the MAG. For comparison, we also mapped the raw reads assigned to *Caroliensis*, which yielded a much higher match score, 38% and 13%, respectively.

Bbmap output examples for **AMX1**:

***Kuenenia*:**

Read 1 data:	pct reads	num reads	pct bases	num bases
mapped:	3.0339%	127	2.7109%	46718

***Jettenia*:**

Read 1 data:	pct reads	num reads	pct bases	num bases
mapped:	2.0301%	58	1.7980%	21487

***Caroliensis*:**

Read 1 data:	pct reads	num reads	pct bases	num bases
mapped:	37.8898%	3868	37.6415%	1445957

Bbmap output examples for **AMX3**:

Kuenenia

Read 1 data:	pct reads	num reads	pct bases	num bases
mapped:	3.9895%	167	4.0966%	70599

Jettenia

mapped:	4.2002%	120	4.5138%	53941
---------	---------	-----	---------	-------

***Caroliensis*:**

mapped:	12.5172%	1277	11.2729%	439379
---------	----------	------	----------	--------

This did not confirm that the majority of the read assignment to *Kuenenia* and *Jettenia* were misinterpreted *Brocadia* sequences. However, matching of the *Kuenenia* and *Jettenia* raw reads to the published *Kuenenia* and *Jettenia* draft genomes in the same manner also resulted only in a low proportion of matches (<1.5%).

Bbmap output for PRJEB22746 (CP049055, *Kuenenia*)

Read 1 data:	pct reads	num reads	pct bases	num bases
mapped:	1.2422%	52	0.9889%	17042
unambiguous:	1.0272%	43	0.8745%	15070
ambiguous:	0.2150%	9	0.1144%	1972
low-Q discards:	0.0000%	0	0.0000%	0

Bbmap output for PRJNA534024 (RHLA01, *Jettenia*)

Read 1 data:	pct reads	num reads	pct bases	num bases
mapped:	0.1400%	40	0.0441%	527
unambiguous:	0.1050%	3	0.0428%	512
ambiguous:	0.0350%	1	0.0013%	15
low-Q discards:	0.0000%	0	0.0000%	0

To investigate this pattern in more detail we further re-ran the taxonomic assignment on the co-assembly used for the binning approach. None of the assembled contigs were assigned to *Kuenenia* or *Jettenia*, and assignment of contigs to Anammox bacteria all referred to the 6 *Brocadia* species in the community. Numbers denote the fraction of reads assigned to the bacterial species.

Candidatus Brocadia;*Candidatus Brocadia caroliniensis*;" 0.130027
Candidatus Brocadia;*Candidatus Brocadia fulgida*;" 0.032002
Candidatus Brocadia;*Candidatus Brocadia sapporoensis*;" 0.065308
Candidatus Brocadia;*Candidatus Brocadia sinica*;" 0.096910
Candidatus Brocadia;*Candidatus Brocadia* sp. UTAMX1;" 0.028322
Candidatus Brocadia;*Candidatus Brocadia* sp. UTAMX2;" 0.025050

This shows that any reads from *Jettenia* or *Kuenenia*, if present, could not be assembled.

In summary, these results confirm the doubts of the reviewer regarding the presence of *Kuenenia* and *Jettenia*, and the accuracy of read based phylogenetic assignment at the species level. Therefore, to avoid misinterpretation of the data and confusion about the AMX diversity in the community we decided to only show the phylum level Kaiju results (Figure 3A) and to remove Figure 3B from the MS. We adapted the text accordingly as well (L324-328). To make the result more transparent, the method used for taxonomic assignment was clarified in the legend of Figure 3 (L843-844)

Figure 4:

Round 1 review

Figure 4 is somewhat strange to me, because it compares unrelated samples in each of the panels. As the reactors were cleaned and reinoculated between temperature treatments, comparing "reactor 3 20°C" to "reactor 3 14°C" is equally meaningful as comparing "reactor 3 20°C" to "reactor 1 14°C" and thus the figure gives a false temperature comparison. I would rather expect that all data be plotted on a single NMDS, or that the figure consists of 2 panels, 1 for 20°C and 1 for 14°C.

Author Response

Thank you for this suggestion but we disagree on this statement. Cleaning and reinoculating was indeed the best strategy to ensure the greatest similarity in starting conditions, because if this had not been done, the experiments on the other temperature would have been performed on a community that had already been in the experimental reactors longer, at a different temperature, and experienced oxygen stress episodes. Our intention was to show that temperature has an influence on the community-wide transcription independent of the reactors. Given the differences between reactors that are also seen in Figure 1, separating the by reactors provides a clearer view. Similar to our findings on the process level we want to show with this figure both the general temperature effect and that the impact of DO has a stronger influence during the 20°C experiment. Both findings are further supported by multiple lines of analysis in the MS and in other figures.

Round 2 review

I'm not sure this is a point where we can agree to disagree, as I am a little worried that this could be unintentional data manipulation. If your effect goes away when including all replicates, it might not be real. I can not think of a type of experiment where replicates are done, and instead of comparing all replicates of one condition with all replicates of the other condition, compare them experiment by experiment (which is what you're doing here). As you have cleaned the reactors in between, conditions the replicates should be independent (i.e. "by reactor" should be meaningless) and you have no justification for comparing them as you do, instead of any other permutation.

I agree that the spread of the points for 14°C experiments is smaller than for the 20°C experiments, indicating a smaller response to the DO perturbation, and expect that this effect is also clearly visible in an NMDS including all samples. Where I disagree is that you can use this visualization to show that the clustering of the points is driven by temperature (rather than by inoculum, for example). A NMDS including all data could clarify this.

I'm worried about this for two reasons:

- 1) if all data points group by experiment, your inoculum effect is larger than your temperature effect, and the figure (and data interpretation) should reflect that.
- 2) if the data clearly group by reactor, there is a batch effect as a result of the systems not being completely identical, and you might need to correct for that.

Thank you very much for your input and the clarifications. We understand your concerns in this matter and of course, we do not want to give the impression of data manipulation. We have therefore re-run the nMDS analysis with the merged datasets

and found the same patterns as displayed in the current figure. The spread of points is much larger in the 20°C experiment, while the points from the 14°C experiment all cluster together. The Figure also shows clearly that the changes in the inoculum are small compared to the differences observed in the 20°C experiment, and that differences between 14°C and 20°C experiments are not related to differences in the inoculum. Therefore, we remain confident in the statement that the impact of DO disturbances seem to have a much larger impact on the 20° global transcriptome in comparison to the 14°C experiment, and that this is not an artefact related to changes in the inoculum.

To make this point clear and address your concerns we have put the full data nMDS in the MS and transferred the analysis separated by reactor from the former figure 4 to the SI. We want to maintain the latter, because this representation is better suited for showing that DO disturbances did not induce the anticipated consistent transcriptional responses. Including the time/stress information on top of temperature and reactor information in the combined nMDS graph would overload the Figure. The manuscript was adjusted accordingly (L400-416):

New Figure 4:

The community-wide transcription differed significantly between the two temperature regimes and significantly between the reactors during the 20°C experiment (PERMANOVA: $p < 0.005$).

The nMDS ordination based on global transcript abundances (Figure 4) displays, independent of the reactor, a tight clustering of the 14°C samples, indicating only minor changes in the transcriptional status of the community in response to the applied DO disturbances. On the other hand, samples from the 20°C experiment are dispersed in the ordination, suggesting a larger transcriptional variance during the 20 °C experiment. The dissimilarities between the two inoculum samples (black triangles), taken one week apart at the exact same time during a SBR cycle, indicate that the inoculum was generally stable, but also that variability in transcriptomic data has to be expected even without experimental intervention – either due to the dynamic nature of this engineered ecosystem or due to methodological error. These differences had no effect on the process level, as we did not observe significant differences in NH₄⁺ removal rates between the two fillings of the experimental reactors.

DO disturbances did not induce a consistent community-wide transcriptional response at either temperature regime, i.e. samples obtained at “stress” conditions did not cluster consistently apart from baseline samples (Supplementary Figure 7).

REVIEWERS' COMMENTS:

Reviewer #1 (Remarks to the Author):

It can be accepted.

Reviewer #2 (Remarks to the Author):

I appreciate your revision from the last round of comments. Overall, I am satisfied with the overall response. This work is great and adds a lot of intellectual merit to the existing literature of anammox.

Reviewer #3 (Remarks to the Author):

I thank the authors for the additional analyses they did in response to my previous feedback, and am fully satisfied with the changes to the manuscript.